# A multiethnic genome-wide association study of primary open-angle glaucoma identifies novel risk loci

Hélène Choquet [1], Seyyedhassan Paylakhi[2], Stephen C. Kneeland[3], Khanh K. Thai[1], Thomas J. Hoffmann[4,5], Jie Yin[1], Mark N. Kvale[4], Yambazi Banda[4], Nicholas G. Tolman[3], Pete A. Williams [3], Catherine Schaefer[1], Ronald B. Melles[6], Neil Risch[1,4,5], Simon W.M. John[3], K. Saidas Nair[2,7] & Eric Jorgenson [1]

Primary open-angle glaucoma (POAG) is a leading cause of irreversible vision loss, yet much of the genetic risk remains unaccounted for, especially in African-Americans who have a higher risk for developing POAG. We conduct a multiethnic genome-wide association study (GWAS) of POAG in the GERA cohort, with replication in the UK Biobank (UKB), and vice versa, GWAS in UKB with replication in GERA. We identify 24 loci ($P < 5.0 \times 10^{-8}$), including 14 novel, of which 9 replicate (near *FMNL2*, *PDE7B*, *TMTC2*, *IKZF2*, *CADM2*, *DGKG*, *ANKH*, *EXOC2*, and *LMX1B*). Functional studies support intraocular pressure-related influences of *FMNL2* and *LMX1B*, with certain *Lmx1b* mutations causing high IOP and glaucoma resembling POAG in mice. The newly identified loci increase the proportion of variance explained in each GERA race/ethnicity group, with the largest gain in African-Americans (0.5–3.1%). A meta-analysis combining GERA and UKB identifies 24 additional loci. Our study provides important insights into glaucoma pathogenesis.

[1] Division of Research, Kaiser Permanente Northern California (KPNC), Oakland, CA 94612, USA. [2] Department of Ophthalmology, School of Medicine, University of California San Francisco (UCSF), San Francisco, CA 94143, USA. [3] The Jackson Laboratory, Howard Hughes Medical Institute, Bar Harbor, ME 04609, USA. [4] Institute for Human Genetics, UCSF, San Francisco, CA 94143, USA. [5] Department of Epidemiology and Biostatistics, UCSF, San Francisco, CA 94158, USA. [6] Department of Ophthalmology, KPNC, Redwood City, CA 94063, USA. [7] Department of Anatomy, School of Medicine, UCSF, San Francisco, CA 94143, USA. Correspondence and requests for materials should be addressed to K.S.N. (email: Saidas.Nair@ucsf.edu) or to E.J. (email: Eric.Jorgenson@kp.org)

G laucoma is a progressive optic neuropathy caused by a combination of genetic and environmental factors and is the leading cause of irreversible blindness worldwide[1]. It is a heterogenous group of diseases, unified by a loss of retinal ganglion cells (RGCs) and the degeneration of their axons in the optic nerve, leading to associated visual field defects. Primary open-angle glaucoma (POAG) is the most common form of glaucoma[2] and is often associated with elevated intraocular pressure (IOP), although up to one-third of POAG patients have IOP levels in the normal range[3]. In addition to high IOP, major risk factors for POAG include advancing age, myopia, race/ethnicity, and positive family history[2,4].

Characterizing the genetic factors influencing an individual's susceptibility to POAG is an important step toward understanding its etiology. A twin study, published in 1987, estimated the heritability of POAG at 0.13[5]. More recently, two studies based on genome-wide array data of unrelated individuals estimated the additive heritability of POAG to range from 24 to 42%, but with inconsistent results between the two studies[6,7]. Genome-wide association studies (GWASs) have reported 17 loci associated with POAG at genome-wide significance, and an additional 2 loci at a suggestive level of significance ($P < 10^{-6}$)[8–18]. Together these loci explain only a small proportion of the genetic contribution to POAG risk, and although most of the reported associations have been validated in an independent study, only a few have been investigated in functional studies (e.g., *CDKN2B-AS*, *SIX6*, and *CAV1/2*)[19–22]. There is also a lack of studies investigating genetic risk factors influencing POAG susceptibility in African ancestry populations and Hispanic/Latino individuals, with only one GWAS published to date[14]. This is a particularly important gap in our understanding of glaucoma etiology, because of the higher prevalence of POAG in these populations (https://nei.nih.gov/health/glaucoma/glaucoma_facts).

To address these gaps, we undertake a GWAS of POAG and an analysis of genetic ancestry in the large and ethnically diverse Genetic Epidemiology Research in Adult Health and Aging (GERA) cohort, consisting of 4 986 POAG cases and 58 426 controls from four race/ethnicity groups (non-Hispanic whites, Hispanic/Latinos, East Asians, and African-Americans). We test novel loci discovered in the current study in an independent external replication cohort: the multiethnic UK Biobank (UKB)[23], which includes 7 329 glaucoma cases (subtype unspecified) and 169 561 controls. Conversely, we then investigate novel glaucoma-associated loci from the UKB in GERA, and conduct a multiethnic meta-analysis combining GERA and UKB. We also examine the effect of newly and previously identified POAG loci on the ancestry effects observed in GERA. In addition to undertaking in silico analyses of associated loci, we examine changes in the expression of candidate genes in RGCs and optic nerve head tissue isolated from a mouse model of glaucoma as a mean of identifying genes that potentially contribute to glaucomatous neurodegeneration. Next, we perform functional characterization of *FMNL2* and *LMX1B*, two of our novel associated loci. First, employing an small interfering RNA (siRNA)-mediated gene silencing of *FMNL2*, we determine a role for *FMNL2* in supporting the function of human trabecular meshwork (HTM) cells, a critical cell type involved in the regulation of IOP. Second, utilizing mutant mouse strains, we demonstrate support for a role of *LMX1B* in the pathogenesis of ocular hypertension (OHTN) and glaucoma in the general population. The current study identifies a total of nine novel loci associated with the risk of POAG or glaucoma at genome-wide significance that replicated in an independent sample, and demonstrates relevant function of *FMNL2* and *LMX1B* using cell line and mouse experiments.

## Results

**GERA cohort**. We conducted the primary discovery analysis in 4 986 POAG cases and 58 426 controls from 4 race/ethnicity groups (non-Hispanic whites, Hispanic/Latinos, East Asians, and African-Americans) in GERA (Table 1). Among GERA subjects, cases were more likely to be older and male. We also observed a much higher prevalence of POAG among African-Americans compared to non-Hispanic whites (16.1% vs. 7.4%), along with a less-pronounced higher prevalence in East Asians (9.9%) and Hispanic/Latinos (7.9%)

Because of the observed variation in POAG prevalence, we examined POAG prevalence in the context of genetic ancestry within each of the four race/ethnicity groups (Fig. 1, Supplementary Table 1). In the African-American group, we observed a significant association ($P = 0.01$) between increasing POAG risk and greater African (compared to European) ancestry (PC1). In the East Asian group, we observed a greater risk of POAG with northern (vs. southern) East Asian ancestry ($P = 7.3 \times 10^{-8}$) (PC2). In the Hispanic/Latino group, we observed higher risk of POAG with greater Native American (vs. European) ancestry ($P = 6.3 \times 10^{-8}$) (PC1).

**Glaucoma loci in GERA**. In our discovery GWAS analysis, we identified 12 independent genome-wide significant ($P < 5 \times 10^{-8}$) loci associated with POAG in the multiethnic meta-analysis ($\lambda = 1.056$ and $\lambda_{1000} = 1.006$, which is reasonable for a sample of this size under the assumption of polygenic inheritance[24]) (Table 2, and Supplementary Figure 1, Supplementary Figure 2). Of the 12 loci, 5 were novel (41.7%), including rs56117902 in *FMNL2* (odds ratio (OR) = 0.88, $P = 1.0 \times 10^{-8}$), rs9494457 in *PDE7B* (OR = 1.16, $P = 3.0 \times 10^{-11}$), rs149154973 near *ELN* (OR = 0.70, $P = 3.9 \times 10^{-9}$), rs324794 near *TMTC2* (OR = 0.87, $P = 6.1 \times 10^{-10}$), and rs2593221 in *TCF12* (OR = 0.86, $P = 8.5 \times 10^{-10}$) (Supplementary Figure 3). We also examined the association of the lead single-nucleotide polymorphisms (SNPs) at the 12 genome-wide significant loci with POAG in each individual race/ethnicity group (Supplementary Table 2 and Supplementary Figure 1). In African-Americans, SNPs rs56117902 at *FMNL2* and rs149154973 near *ELN* were both nominally associated with POAG ($P = 0.017$ and 0.047 for rs56117902 and rs149154973, respectively). In Hispanic/Latinos, we found a suggestive association between *PDE7B* rs9494457 and POAG risk ($P = 0.005$). In East Asians, we detected a nominal association of POAG with *TCF12* rs2593221 ($P = 0.019$). Except for rs149154973 near *ELN*, all SNPs at novel loci showed consistent direction of effects across all race/ethnicity groups, and no significant heterogeneity was observed between race/ethnicity groups.

**Replication in an independent external cohort**. We then tested the five lead SNPs representing each of the five novel loci for replication in an independent, external meta-analysis in the multiethnic UKB[23], which includes 7 329 glaucoma cases and 169 561 controls from five race/ethnicity groups (European, East Asian, South Asian, African British, and mixed ancestries) (Supplementary Table 3). Three SNPs, including rs56117902 at *FMNL2*, rs9494457 at *PDE7B*, and rs324794 near *TMTC2*, replicated at Bonferroni significance ($P = 7.9 \times 10^{-5}$, $7.2 \times 10^{-6}$, and $6.8 \times 10^{-3}$, respectively) with the same direction of effect between the discovery association and replication sample (Table 2). In addition, rs2593221 near *TCF12* was associated at a nominal level ($P = 0.025$). No data were available in UKB to test SNP rs149154973 for replication.

**Conditional and epistasis analyses**. We next searched for additional genome-wide significant SNPs within a 2 Mb window of

**Table 1 Characteristics of the POAG cases and controls from GERA cohort**

| | | Controls | Cases |
|---|---|---|---|
| Total | N (proportion that are cases) | 58 426 | 4 986 (7.9%) |
| Age (years) | Mean ± SD | 69.7 ± 13.0 | 79.6 ± 9.4 |
| Sex | Women - N (proportion that are cases) | 35 193 | 2 689 (7.1%) |
| | Men - N (proportion that are cases) | 23 233 | 2 297 (9.0%) |
| NTG cases | N | — | 743 |
| Controls with an OHTN diagnosis | N | 881 | — |
| Median IOP | Mean ± SD | 14.9 ± 2.6 | 15.5 ± 2.9 |
| Non-Hispanic white | N (proportion that are cases) | 48 065 | 3 836 (7.4%) |
| | Age (years), mean ± SD | 70.7 ± 12.5 | 80.5 ± 9.0 |
| | Sex, number of women/men | 28 798/19 266 | 2 115/1 721 |
| Hispanic/Latino | N (proportion that are cases) | 4 778 | 411 (7.9%) |
| | Age (years) Mean ± SD | 65.0 ± 14.4 | 75.5 ± 10.1 |
| | Sex, number of Women / Men | 2 965/1 813 | 218/193 |
| East Asian | N (proportion that are cases) | 4 034 | 441 (9.9%) |
| | Age (years), mean ± SD | 64.6 ± 14.4 | 76.6 ± 10.0 |
| | Sex, number of women/men | 2 453/1 581 | 202/239 |
| African-American | N (proportion that are cases) | 1 549 | 298 (16.1%) |
| | Age (years), mean ± SD | 66.7 ± 13.3 | 77.7 ± 9.3 |
| | Sex, number of women/men | 976/573 | 154/144 |

*N* number, *SD* standard deviation, *POAG* primary open-angle glaucoma, *NTG* normal tension glaucoma, *OHTN* ocular hypertension, defined as participants who had a diagnosis of ocular hypertension but have no diagnosis of any type of glaucoma, *IOP* intraocular pressure, *Age* age at last vision exam

the 12 lead SNPs (±1.0 Mb with respect to the lead SNP), including those 12 lead SNPs identified in the discovery GERA multiethnic GWAS analysis as covariates. We identified 3 additional SNPs with Bonferroni-level evidence of association with POAG that appeared to be independent of the lead SNP signals ($P < 0.0001$) at 3 loci (*UCK2* on chromosome 1 (near *TMCO1*), *CDKN2B-AS1* on chromosome 9, and near *SIX1* on chromosome 14) (Supplementary Table 4). We then conducted an epistasis analysis of all pairs of lead SNPs in each race/ethnicity group, but did not observe significant epistatic interactions between SNPs after Bonferroni correction ($P < 1.9 \times 10^{-4} = 0.05/(((12 \times 11)/2) \times 4)))$.

**Secondary and subgroup analyses**. To further investigate whether the POAG susceptibility loci identified in this study influence glaucoma susceptibility through their effect on IOP or independently of IOP, we conducted three additional analyses. First, we examined the association of lead SNPs at these loci with IOP, which was assessed in the GERA cohort (see Methods). Of our 12 lead SNPs, 6 were associated at a Bonferroni level of significance ($P < 0.0014$ for 12 SNPs in 3 analyses): rs7524755 at *TMCO1*; rs56117902 at *FMNL2*; rs59521811 at *AFAP1*; rs9494457 at *PDE7B*; rs2472493 near *ABCA1*; and rs9913911 at *GAS7* (Supplementary Table 5). Each of the POAG risk alleles was associated with greater IOP.

We then examined the association of these SNPs in two POAG subgroups: subjects with normal tension glaucoma (NTG) compared to controls (743 NTG cases vs. 58 426 controls) and subjects with high tension glaucoma (HTG) compared to controls (4243 HTG cases vs. 58 426 controls). In the NTG analysis, of our 12 lead SNPs, 3 were associated at a Bonferroni level of significance and each had more extreme odds ratios in comparison to their association in the POAG GWAS: rs6913530 near *CDKN1A*; rs10811645 at *CDKN2B-AS1*; and rs35155027 in *SIX1-SIX6* (Supplementary Table 5). In the HTG analysis, all 12 lead SNPs were associated at Bonferroni significance, consistent with this subset making up the majority of the sample.

These results suggest that some of the POAG risk variants act through their effect on IOP, while others do not. These findings are consistent with previous studies reporting NTG associations at *CDKN2B-AS1*[11,17]. However, we did not confirm previous NTG associations on 8q22 (top SNP rs284491[C], OR = 1.06, $P = 0.29$) or 12q (rs2041895[C], OR = 1.04, $P = 0.43$)[11,17] (Supplementary Table 6).

**Replication of previous POAG GWAS results**. We also investigated in GERA the lead SNPs within 19 loci associated with POAG at a genome-wide significance level or showing suggestive evidence ($P < 10^{-6}$) in previous studies[8–18] (Supplementary Table 7). Six of the 19 replicated at a genome-wide level of significance in our GERA multiethnic meta-analysis (including *TMCO1* rs4656461, *CDKN1A* rs67530707, *CDKN2B-AS1* rs4977756, *ABCA1* rs2472493, *SIX1/SIX6* rs10483727, and *GAS7* rs9897123) (Supplementary Table 7). Further, 7 additional SNPs replicated at Bonferroni significance ($P < 0.05/19 = 0.0026$), and 1 (*ATXN2* rs7137828) showed nominal evidence of association ($P = 0.0034$). The previously reported[14] rs199748604 (*TRIM9-TMX1*) reached nominal significance in African-Americans, but not in the other race/ethnicity groups.

In contrast, two SNPs (at *NCKAP5* and *PMM2*), which were reported with suggestive evidence of association with POAG in previous studies of Asian individuals[10,12], were associated with POAG in neither the current GERA multiethnic meta-analysis ($P > 0.05$), nor in the East Asian race/ethnicity group. The absence of replication in our study is unlikely to be due to a lack of statistical power, as we have estimated that our study of 4986 cases and 58 426 controls (full GERA sample) or 441 cases and 4034 controls (GERA East Asian race/ethnicity group) had more than 80% power to detect the previously reported effects for both SNPs rs7588567 and rs3785176. We note that in our GERA sample, *NCKAP5* rs7588567 had a moderate imputation quality score $r^2$ in non-Hispanic whites ($r^2 = 0.64$), as well as in Hispanic/Latinos ($r^2 = 0.63$) and in African-Americans ($r^2 = 0.61$). For this reason, rs7588567 was excluded from the meta-analysis across the 4 race/ethnicity

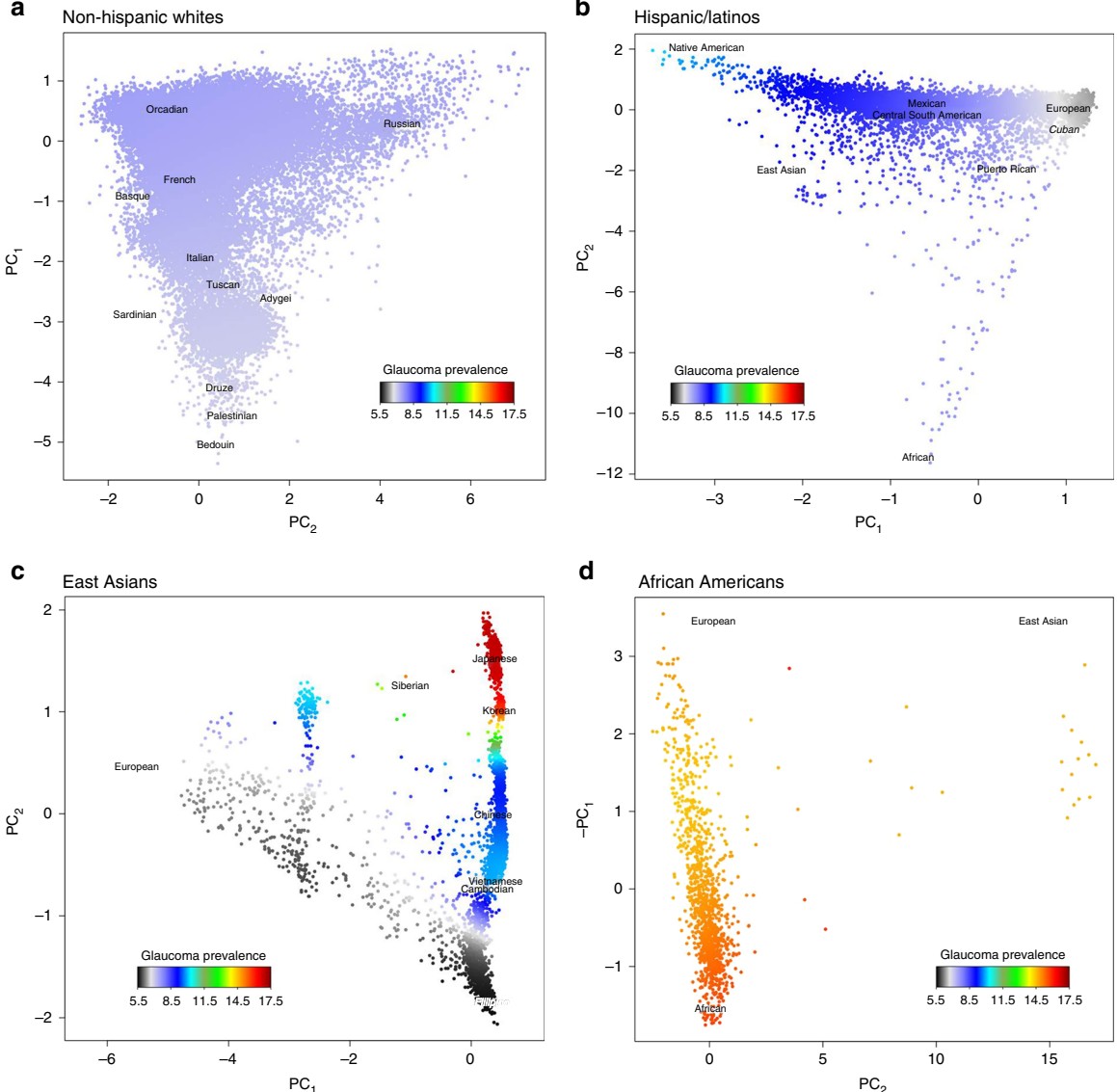

**Fig. 1** Plots of POAG prevalence versus genetic ancestry in GERA. POAG prevalence is indicated on a color scale, standardized across groups, with warmer colors indicating higher prevalence. Axes reflect the first two principal components of ancestry in each group. The phenotype distribution was smoothed over the PCs (within the individuals in each respective figure), which were divided by their standard deviation for interpretability (see Methods). Human Genome Diversity Project populations are plotted at their relative positions in each figure. Human Genome Diversity Project populations are presented in a plain font, and GERA populations are presented in italics font. **a** Non-Hispanic whites, **b** Hispanic/Latinos, **c** East Asians, and **d** African-Americans

groups. Consistent with a previous study conducted in African-American women[14], *DNAJC24* rs542340 and *RBFOX1* rs192917960 were observed at a low frequency in African-Americans, and were not polymorphic (minor allele frequency (MAF) < 0.01%) in other race/ethnicity groups. So, these two SNPs were excluded from the GERA multiethnic meta-analysis and were not associated with POAG in the African-American race/ethnicity group.

**Array heritability estimates for POAG risk in GERA**. We then estimated the proportion of variance in POAG risk explained by all SNPs on the genotyping array in the GERA non-Hispanic white sample using GCTA[25]. Assuming a prevalence of 3% for POAG[2], we estimated an overall heritability of 0.26 (s.e. = 0.01). Our results are consistent with a recent phenome-wide heritability study conducted in the UKB and showing an overall heritability estimate of 0.26 (s.e. = 0.06) for glaucoma[7].

**Variability in POAG risk explained**. We then examined the proportion of variance explained by previously known and newly discovered SNPs in GERA. Previously reported SNPs explained 2.1% of the variation in POAG risk in non-Hispanic whites, 0.5% in African-Americans, 2.0% in Hispanic/Latinos, and 0.3% in East Asians. Newly identified variants increased the proportion of variance explained to 3.0% in non-Hispanic whites, 3.1% in African-Americans, 3.3% in Hispanic/Latinos, and 0.5% in East Asians.

**Lead glaucoma SNPs from UKB and replication in GERA**. A genome-wide analysis of glaucoma in the UKB sample identified 18 glaucoma-associated loci that achieved a genome-wide level of significance, of which 9 have not been previously reported to be associated with glaucoma (Table 3 and Supplementary Figure 2). We then investigated the 9 lead SNPs representing each of the 9 independent novel loci for replication in the GERA POAG

**Table 2 Lead genome-wide significant SNP for each independent locus identified in the GERA discovery GWAS of POAG**

| SNP | Chr | Pos | Locus | Alleles | GERA discovery cohort | | UKB replication cohort | |
|---|---|---|---|---|---|---|---|---|
| | | | | | OR (95% CI) | P | OR (95% CI) | P |
| rs7524755 | 1 | 165694897 | *TMCO1* | T/C | 1.35 (1.27–1.44) | $1.6 \times 10^{-21}$ | 1.37 (1.31–1.43) | $8.3 \times 10^{-41}$ |
| rs56117902 | 2 | 153304730 | *FMNL2* | A/C | 0.88 (0.84–0.92) | $1.0 \times 10^{-8}$ | 0.93 (0.90–0.97) | $7.3 \times 10^{-5}$ |
| rs59521811 | 4 | 7909772 | *AFAP1* | T/C | 0.86 (0.82–0.90) | $1.2 \times 10^{-11}$ | 0.86 (0.83–0.89) | $9.8 \times 10^{-17}$ |
| rs6913530 | 6 | 36598209 | *near CDKN1A* | G/A | 0.86 (0.82–0.90) | $2.3 \times 10^{-9}$ | 0.98 (0.95–1.02) | 0.44 |
| rs9494457 | 6 | 136474794 | *PDE7B* | T/A | 1.16 (1.11–1.22) | $3.0 \times 10^{-11}$ | 1.08 (1.05–1.12) | $7.4 \times 10^{-6}$ |
| rs149154973 | 7 | 73322211 | *near ELN* | TG/T | 0.70 (0.62–0.79) | $3.9 \times 10^{-9}$ | NA | NA |
| rs10811645 | 9 | 22049656 | *CDKN2B-AS1* | G/A | 0.77 (0.74–0.80) | $2.8 \times 10^{-32}$ | 0.86 (0.83–0.89) | $1.0 \times 10^{-18}$ |
| rs2472493 | 9 | 107695848 | *near ABCA1* | G/A | 1.17 (1.12–1.22) | $1.2 \times 10^{-13}$ | 1.15 (1.12–1.19) | $6.3 \times 10^{-17}$ |
| rs324794 | 12 | 83946450 | *near TMTC2* | G/T | 0.87 (0.83–0.91) | $6.1 \times 10^{-10}$ | 0.95 (0.92–0.99) | $6.8 \times 10^{-3}$ |
| rs35155027 | 14 | 61095174 | *SIX1/SIX6* | G/C | 1.17 (1.12–1.23) | $1.1 \times 10^{-12}$ | 1.13 (1.10–1.17) | $6.2 \times 10^{-13}$ |
| rs2593221 | 15 | 57501414 | *TCF12* | A/G | 0.86 (0.82–0.90) | $8.5 \times 10^{-10}$ | 0.96 (0.92–0.99) | 0.025 |
| rs9913911 | 17 | 10031183 | *GAS7* | A/G | 1.22 (1.17–1.28) | $1.5 \times 10^{-18}$ | 1.17 (1.13–1.21) | $2.2 \times 10^{-17}$ |

*P*-values in bold achieved genome-wide level of significance ($P < 5 \times 10^{-8}$); loci that are underlined are novel
*SNP* single-nucleotide polymorphism, *Chr* chromosome, *Pos* position, *OR* odds ratio, *NA* not available

**Table 3 Lead genome-wide significant SNP for each independent locus identified in the UK Biobank and replication in GERA**

| SNP | Chr | Pos | Locus | Alleles | UKB discovery cohort | | GERA replication cohort | |
|---|---|---|---|---|---|---|---|---|
| | | | | | OR (95% CI) | P | OR (95% CI) | P |
| rs2814471 | 1 | 165739598 | *TMCO1* | C/T | 1.39 (1.32–1.45) | $7.5 \times 10^{-44}$ | 1.34 (1.26–1.43) | $2.0 \times 10^{-20}$ |
| rs56335522 | 2 | 213758234 | near *IKZF2* | G/C | 1.18 (1.11–1.25) | $1.7 \times 10^{-8}$ | 1.20 (1.11–1.29) | $2.0 \times 10^{-6}$ |
| rs34201102 | 3 | 85137499 | *CADM2* | A/G | 1.11 (1.08–1.15) | $6.1 \times 10^{-10}$ | 1.10 (1.05–1.15) | $2.1 \times 10^{-5}$ |
| rs9853115 | 3 | 186131600 | *near DGKG* | T/A | 1.11 (1.07–1.15) | $1.7 \times 10^{-9}$ | 1.10 (1.05–1.14) | $2.2 \times 10^{-5}$ |
| rs9330348 | 4 | 7883887 | *AFAP1* | C/G | 1.16 (1.12–1.20) | $2.4 \times 10^{-18}$ | 1.15 (1.10–1.20) | $5.7 \times 10^{-10}$ |
| rs76325372 | 5 | 14837332 | *ANKH* | A/C | 1.14 (1.09–1.18) | $1.4 \times 10^{-10}$ | 1.09 (1.04–1.14) | 0.00045 |
| rs2073006 | 6 | 637465 | *EXOC2* | C/T | 0.86 (0.82–0.90) | $2.8 \times 10^{-10}$ | 0.86 (0.81–0.92) | $1.2 \times 10^{-6}$ |
| rs12699251 | 7 | 11679113 | *THSD7A* | A/G | 0.90 (0.87–0.93) | $1.5 \times 10^{-9}$ | 0.95 (0.90–0.99) | 0.017 |
| rs6969706 | 7 | 116154831 | *CAV1/CAV2* | G/T | 0.90 (0.86–0.93) | $8.4 \times 10^{-9}$ | 0.91 (0.87–0.95) | $9.6 \times 10^{-5}$ |
| rs2514884 | 8 | 108276873 | *ANGPT1* | C/T | 0.84 (0.80–0.88) | $1.4 \times 10^{-11}$ | 0.93 (0.87–0.98) | 0.014 |
| rs1333037 | 9 | 22040765 | *CDKN2B-AS1* | C/T | 0.84 (0.81–0.87) | $2.4 \times 10^{-23}$ | 0.77 (0.74–0.81) | $2.9 \times 10^{-29}$ |
| rs2472493 | 9 | 107695848 | near *ABCA1* | G/A | 1.15 (1.12–1.19) | $6.3 \times 10^{-17}$ | 1.17 (1.12–1.22) | $1.2 \times 10^{-13}$ |
| rs55770306 | 9 | 129388033 | *LMX1B* | C/A | 0.86 (0.83–0.90) | $5.5 \times 10^{-14}$ | 0.90 (0.85–0.95) | 0.00019 |
| rs2274224 | 10 | 96039597 | *PLCE1* | G/C | 0.91 (0.88–0.94) | $3.1 \times 10^{-8}$ | 0.96 (0.92–1.00) | 0.058 |
| rs12806740 | 11 | 120203628 | *TMEM136* | G/A | 0.90 (0.87–0.93) | $3.8 \times 10^{-9}$ | 0.93 (0.89–0.97) | 0.0017 |
| rs34935520 | 14 | 61091401 | *SIX1/SIX6* | G/A | 1.14 (1.10–1.17) | $3.6 \times 10^{-13}$ | 1.17 (1.12–1.22) | $1.8 \times 10^{-12}$ |
| rs9913911 | 17 | 10031183 | *GAS7* | A/G | 1.17 (1.13–1.21) | $2.2 \times 10^{-17}$ | 1.22 (1.17–1.28) | $1.5 \times 10^{-18}$ |
| rs58714937 | 22 | 19856710 | near *TXNRD2* | C/T | 1.15 (1.10–1.21) | $8.3 \times 10^{-9}$ | 1.15 (1.09–1.23) | $2.6 \times 10^{-6}$ |

*P*-values in bold achieved genome-wide level of significance ($P < 5 \times 10^{-8}$); loci that are underlined are novel
*SNP* single-nucleotide polymorphism, *Chr* chromosome, *Pos* position, *OR* odds ratio

analysis. Of the 9 novel glaucoma-associated SNPs, 6 replicated at Bonferroni significance ($P < 0.0055 = 0.05/9$) in the GERA meta-analysis. These include rs56335522 near *IKZF2* ($P = 2.0 \times 10^{-6}$), rs34201102 in *CADM2* ($P = 2.1 \times 10^{-5}$), rs9853115 near *DGKG* ($P = 2.2 \times 10^{-5}$), rs76325372 in *ANKH* ($P = 0.00045$), rs2073006 in *EXOC2* ($P = 1.2 \times 10^{-6}$), and rs55770306 in *LMX1B* ($P = 0.00019$) (Table 3). Two additional SNPs were associated in GERA at a nominal level of significance ($P < 0.05$); all 9 SNPs had the same direction of effect in GERA as in UKB.

**Multiethnic meta-analysis of UKB and GERA.** We then conducted a multiethnic meta-analysis, combining the GERA and UKB cohorts. We identified 47 loci at a genome-wide level of significance, including an additional 24 novel loci (Fig. 2 and Supplementary Table 8). These 24 loci will need to be validated in an external replication sample to confirm their role in glaucoma susceptibility.

**Ancestry analysis adjusting for known risk SNPs.** To determine whether known POAG-associated loci explain the observed

associations of genetic ancestry with the risk of POAG, we repeated the ancestry analysis within GERA Hispanic/Latinos, East Asians, and African-Americans, including all newly and previously identified POAG lead SNPs. The ancestry associations were significantly attenuated in African-Americans, but only slightly in the Hispanic/Latinos and slightly strengthened in East Asians (Supplementary Table 9).

**Gene expression in human ocular tissues.** To prioritize genes for further investigation, we identified the 95% credible set of variants in each locus identified in either GERA or UKB (Supplementary Data 1). We then examined expression levels in silico for the 28 genes in the 24 unique loci that contained associated 95% credible set variants in human ocular tissues using two publicly available databases: the Ocular Tissue Database (OTDB)[26]; and EyeSAGE[27,28]. According to OTDB, most of the identified genes were expressed in most ocular tissues (Supplementary Table 10).

**Gene expression in mouse ocular tissues.** RGCs are the primary cell type affected in glaucoma, however, treatments confering

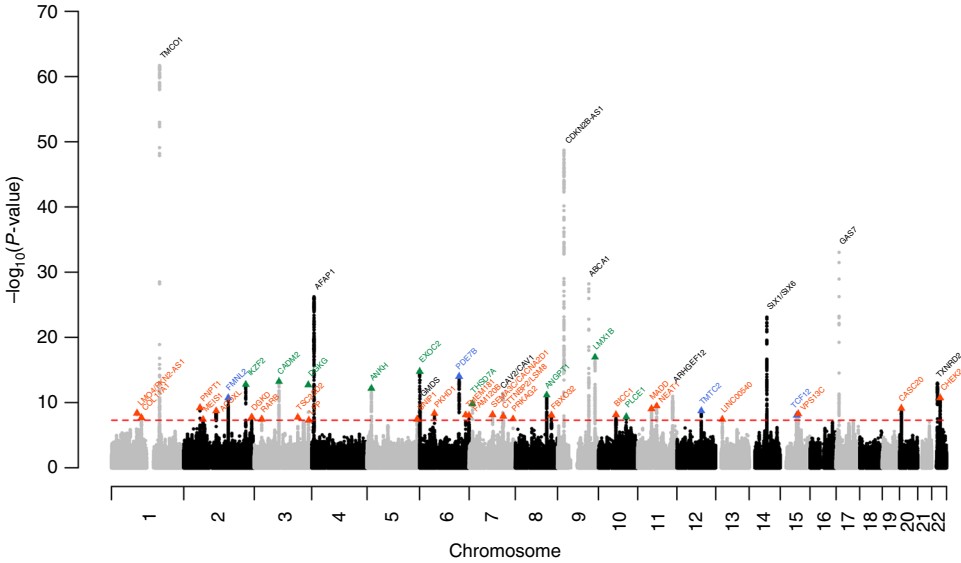

**Fig. 2** Manhattan plot of the GWAS meta-analysis (GERA + UKB) for glaucoma. Association results (−log10 *P*-values) are plotted for each chromosome. Names of loci and lead variants are indicated in color: previously identified loci are in black, blue triangles indicate GERA lead variants at novel loci identified in GERA, green triangles are lead UKB-identified variants at novel loci, and orange triangles are for lead GERA + UKB-identified variants at novel loci

cytoprotection to RGCs have not been identified due to limitations in our understanding of glaucomatous degeneration. Susceptibility factors both intrinsic to RGCs and the cells that surround the RGC axons in the optic nerve head, including astroglia, microglia, and endothelial cells are thought to be critical mediators of the degenerative process. We first determined whether genes in the associated loci serve as susceptibility factors contributing to glaucomatous neurodegeneration. To this end, we assessed the expression of each gene implicated in the 95% credible sets and that also has a mouse homolog in the RGCs or the optic nerve head tissue of the DBA/2J (D2) inherited mouse model of glaucoma (Supplementary Data 1—see Methods). Specifically, we utilized previously published RNA-sequencing datasets from RGCs isolated from D2 retina[29], and the microarray gene expression datasets of the D2 optic nerve head, a critical injury site mainly comprising of RGC axons, astroglia, microglia, and blood vessels[30]. These datasets were compared to their respective age- and sex-matched genetic controls (D2-*Gpnmb+*, a strain that does not develop high IOP or glaucoma)[29]. We specifically assessed the expression of 24 genes in 4 distinct glaucoma groups (grouping performed by hierarchical clustering (HC) based on gene expression patterns) or various stages based on disease severity (see Methods) compared to the control group.

Based on expression profiling of the RGCs, 12 of the genes were differentially expressed (at a false discovery rate (FDR) *q* < 0.05) in one or more of the glaucoma groups compared to controls (Supplementary Figure 4a). These included *Angpt1, Ank, Cadm2, Fmnl2, Plce1, Thsd7a,* and *Tmtc2* at the novel POAG/glaucoma loci identified in the current study. Seven of the 12 differentially expressed genes show moderate to high expression in mouse RGCs (i.e., at a normalized count per million of >100), including *Afap1, Ank, Cadm2, Gas7, Thsd7a, Tmem136,* and *Tmtc2* (Supplementary Figure 4b). Based on the gene profiling in the optic nerve head tissue, *Cadm2, Cdkn2b,* and *Tmtc2* exhibit altered expression very early and across all stages of glaucoma (early to severe, Supplementary Table 11). Thus, expression profiling has facilitated identification of initial set of genes that warrant further interrogation to determine their potential role in modulating susceptibility of RGC cells and their axons in response to high IOP.

**Trabecular meshwork modifications due to *FMNL2* silencing.** We hypothesized that *FMNL2*, a gene in one of the novel POAG-associated loci that is also associated with IOP[31], supports trabecular meshwork (TM) function relevant to aqueous humor outflow regulation. FMNL2 belongs to the formin-related family of proteins that acts as a downstream effector of CDC42 (Rho family member) regulating the shape and movement of cells through its effect on actin nucleation and elongation[32,33]. Regulation of actin cytoskeletal dynamics, including formation of actin stress fibers, is critical for contractile properties of cells. The outflow of aqueous humor through the ocular drainage system is dependent on the contractile properties of TM cells[34,35]. Here we examined whether FMNL2 contributes to the assembly of actin stress fibers, and induces cell morphological changes of TM cells, a process critical for supporting contractile activity.

To this end, cells from a HTM cell line were transfected with siRNAs to suppress the expression of *FMNL2*. We confirmed that siRNAs targeting *FMNL2* caused a significant knockdown of *FMNL2* expression, but did not alter the expression of a related gene *FMNL3*, a paralog of *FMNL2* (Fig. 3a). Our results show that serum-starved HTM cells transfected with *FMNL2* siRNA exhibited a refractile morphology, appearing deformed or rounded, whereas cells transfected with control siRNA, exhibited a more flattened or spread morphology (Fig. 3b, c and Supplementary Figure 5). Moreover, in the serum-starved cells, the actin stress fibers, visualized by phalloidin staining of actin filaments, were disrupted in both the control or *FMNL2* siRNA-transfected HTM cells (Fig. 3b and Supplementary Figure 5). Overall, our data suggest that knockdown of *FMNL2* induces changes in HTM cell morphology likely due to its effect on actin stress fiber assembly.

**Genetic background influenced the effect of *Lmx1b* mutations.** Mutations in *LMX1B* are previously established to cause nail-patella syndrome (NPS), a rare developmental disorder characterized by skeletal abnormalities, as well as kidney and eye defects[36,37]. The presentation of glaucoma in NPS patients is variable; some patients develop a phenotype resembling open-angle glaucoma, while in others, glaucoma is accompanied by developmental defects of the ocular anterior segment, including cataracts and abnormalities of the iris and cornea[37,38]. We have

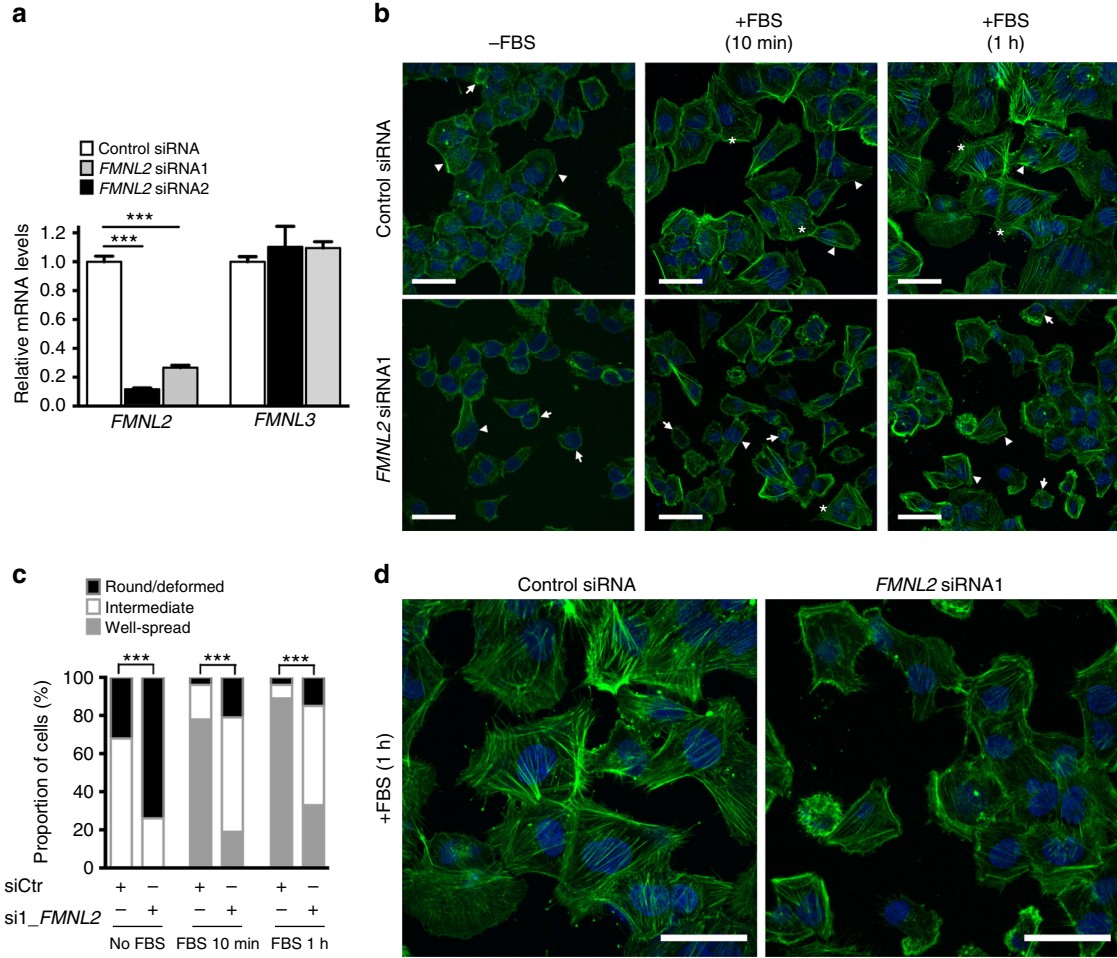

**Fig. 3** *FMNL2* knockdown modifies human trabecular meshwork cell morphology. Following siRNA-mediated knockdown of *FMNL2* for 24 h, HTM cells were serum-starved for 24 h in order to disrupt the normal organization of actin fibers. The cells were then supplemented with fetal bovine serum (FBS) for 10 min or 1 h to induce formation of actin stress fibers and cell spreading. **a** Both the *FMNL2* siRNA1 and -siRNA2 caused significant knockdown of *FMNL2* mRNA as assessed by qPCR. The expression of *FMNL3*, a paralog of *FMNL2* was not altered. **b** Fluorescent microscopic images of phalloidin-FITC-stained F-actin shows that the actin stress fibers are poorly detectable in serum-starved control siRNA-transfected cells (−FBS), and following treatment with FBS (+FBS for 10 min or 1 h) the stress fibers are prominently visible. Consistent with appearance of actin stress fibers, the control siRNA-transfected HTM cells treated with FBS for 10 min are well spread. The actin stress fibers of the *FMNL2*-siRNA1 transfected HTM cells are poorly formed even after 1 h treatment with FBS. Moreover, many of these cells appear deformed/rounded or in general exhibiting poor cell spreading. **c** A bar graph showing proportion of cells (show as percentage) in each of the three groups of cells with distinct morphology (see Methods). In presence of FBS, the percentage of deformed/rounded or modestly spread cells (intermediate) is significantly more in *FMNL2* siRNA1 transfected as compared to control siRNA-transfected HTM cells, which primarily show a well-spread morphology. **d** Magnified fluorescent miscroscopic images of phalloidin-FITC-stained F-actin presented in **b**. White arrows indicate cells with deformed/rounded morphology, arrowheads indicate modestly spread cells (intermediate), and asterisks indicate well-spread cells. Results are mean ± s.e.m. of three independent experiments. Significance of difference was determined by unpaired two-tailed *t*-test and Pearson' Chi-squared test for **a** and **c**, respectively. ***$P < 0.005$. Scale bars = 50 μm

previously characterized a mouse model carrying a dominant missense mutation in *Lmx1b* (*Lmx1b*$^{V265D}$) that develops elevated IOP and glaucoma along with severe ocular developmental anomalies[39]. Here, to functionally evaluate whether *Lmx1b* mutations can cause glaucoma without obvious developmental anomalies, we studied the effects of the same mutation on two different genetic backgrounds, and a new mutation that has not previously been functionally characterized. Specifically, the *Lmx1b*$^{V265D}$ allele was backcrossed onto the C57BL/6J (B6) and 129S6/SvEvTac (129) strain backgrounds; the *Lmx1b*$^{Q82X}$ allele was present on a DBA/2J.Gpnmb(wt) (D2) background.

Slit lamp-based clinical eye examination showed that the B6 mutant mice exhibit a severe developmental phenotype characterized by malformed eccentric pupils, irido-corneal strands, corneal haze, and corneal scleralization (Fig. 4a). The

developmental phenotype on a 129 background was much milder, and mainly limited to mild pupillary abnormalities in about half of the mice. No major developmental abnormalities were detected in D2 mice with the *Lmx1b*$^{Q82X}$ allele (focal corneal keratopathy is an unrelated strain characteristic of D2 mice that is frequently present in both wild-type (WT) and mutant mice). Further, with age, high IOP often results in a more dilated pupil in D2.*Lmx1b*$^{Q82X/+}$ mice. The B6.*Lmx1b*$^{V265D/+}$ and D2. *Lmx1b*$^{Q28X/+}$ mice were highly susceptible to developing glaucomatous nerve damage (Fig. 4b, c). Finally, *Lmx1b* mutations induced elevated IOP in all three strain backgrounds (Fig. 4d, e). IOP is highly variable in the B6.*Lmx1b*$^{V265D/+}$, which is possibly caused by a variety of reasons. One possible explanation for the spread of IOPs is due to *Lmx1b* mutants exhibiting abnormal corneas that are often stretched and

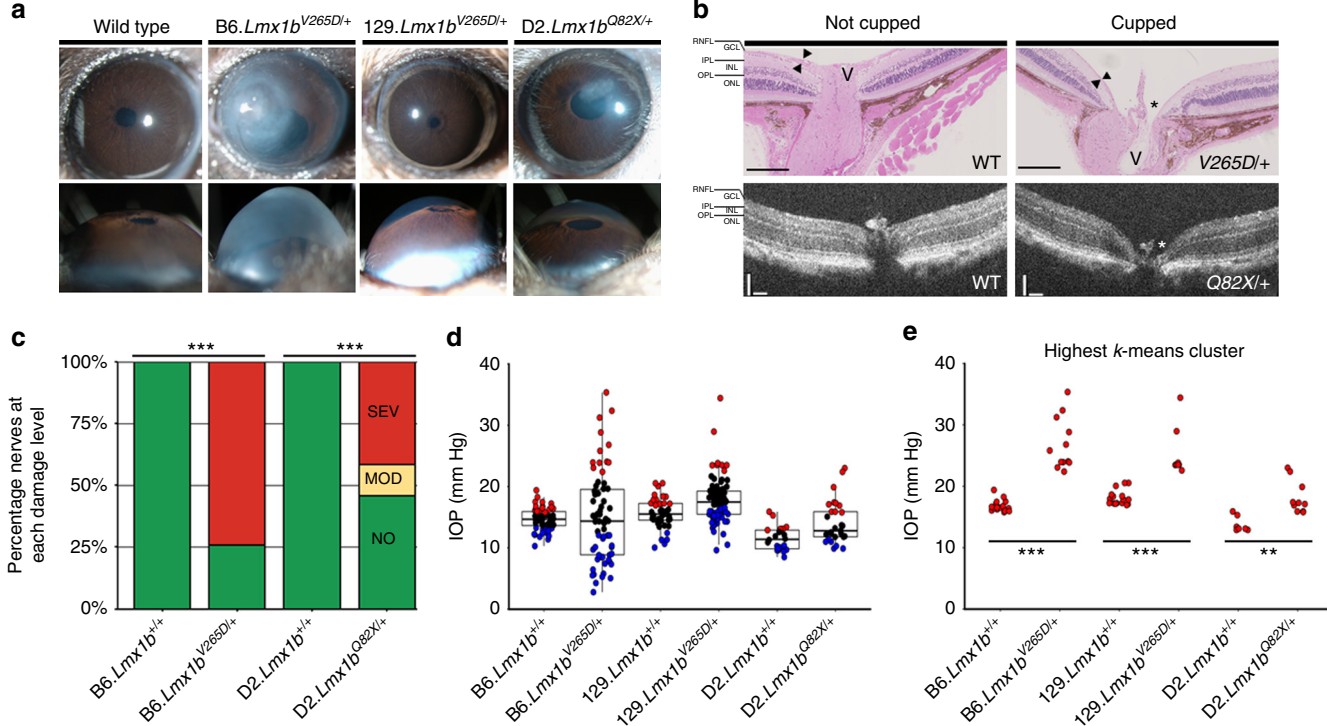

**Fig. 4** Lmx1b can induce high IOP and glaucoma in developmentally normal eyes. **a** Slit lamp examination of control B6.$Lmx1b^{+/+}$, B6.$Lmx1b^{V265D/+}$, 129.$Lmx1b^{V265D/+}$, and D2.$Lmx1b^{Q82X/+}$ at 9 months of age. All D2 mice were wild-type (WT) for Gpnmb. Lmx1b WT eyes on all backgrounds were developmentally normal and so only the B6 controls are shown. B6 mutant mice have a severe developmental phenotype, involving severely malformed eccentric pupils, irido-corneal strands, corneal haze, and corneal scleralization (upper panels). These developmental phenotypes are much milder or absent in mutants on the other backgrounds, with about 50% of mutants on the 129 background having very mild pupillary abnormalities. No mutants on the D2 background had developmental abnormalities. In WT eyes, we can distinguish a very shallow anterior chamber, and, as a result of IOP elevation, the chamber depth increases to different degrees in mutants on each genetic background (lower panels). $N \geq 30$ for each group. **b** Optic nerve cupping (asterisks) and retinal nerve fiber layer (RNFL) thinning (arrowheads), two hallmarks of glaucoma, was present in mutant eyes with severely affected optic nerves but not in WT eyes (10 months' B6 H&E-stained sections, top panels, and 13 months' D2 eyes SD-OCT images shown, bottom panels). Scale bar = 200 μm for H&E-stained images and 100 μm for SD-OCT images. V optic vessel, GCL ganglion cell layer, IPL inner plexiform layer, INL inner nuclear layer, OPL outer plexiform layer, ONL outer nuclear layer. **c** Nerve damage plots for 10-month B6 and 14.5-month-old D2 eyes $N \geq 24$ for each group, Fisher's exact test: ***$P < 0.001$. **d** A proportion of mice carrying the Lmx1b mutations exhibit elevated IOP on all three strain backgrounds as compared to their respective controls. IOP values are clustered within each group using $k$-means clustering ($k = 3$), blue = low IOP cluster, black = average IOP cluster, red = high IOP cluster. $N \geq 24$ for each group, 6 months old shown. **e** IOP values in the high IOP clusters are significantly elevated in Lmx1b mutant mice compared with WT controls. Statistical analysis was done using a one-way ANOVA followed by Tukey's HSD, ***$P < 0.001$, **$P < 0.01$

sometimes perforate. This is a likely explanation, especially considering that B6 Lmx1b mutants have the most severely affected corneas as well as the lowest IOPs. Additionally, variability in IOP may result from stochastic variation in developmental consequences resulting from the Lmx1b mutation. As precedents, various mutant genes that affect development of ocular drainage structures also variably result in maldevelopment of the ciliary body[40,41]. Overall, our data demonstrate that genetic background has a profound impact on the manifestation of glaucoma-related phenotypes caused by mutations in Lmx1b. Importantly, the presence of high IOP or glaucoma in eyes without major developmental anomalies further supports a role of LMX1B in the pathogenesis of OHTN and glaucoma in the general population.

## Discussion

In the large, ethnically diverse GERA cohort, we discovered 5 novel genome-wide significant POAG loci, of which 3 (FMNL2, PDE7B, and near TMTC2) replicated at a Bonferroni level of significance in the UKB cohort. We also investigated 9 novel glaucoma-associated loci from UKB in GERA, and 6 of the novel

loci replicated at Bonferroni significance (near IKZF2, CADM2, near DGKG, ANKH, EXOC2, and LMX1B). Further, a multiethnic meta-analysis combining GERA and UKB identified an additional 24 novel loci that await validation in an external replication cohort. Our results also confirmed the association of variants with POAG in 14 previously reported loci, including TGFBR3, TMCO1, AFAP1, FOXC1, GMDS, CDKN1A, CAV1/CAV2, CDKN2B-AS1, ABCA1, ARHGEF12, ATXN2, SIX1/SIX6, GAS7, and TXNRD2.

We recently reported[31] that variants at FMNL2 are associated with variation in IOP levels. Here we have demonstrated that variants at FMNL2 are also associated with the risk of POAG, with the higher IOP level allele associated with an increased risk of POAG. Therefore, FMNL2 likely contributes to glaucoma susceptibility through its effect on elevated IOP. Consistent with this suggested mechanism of action, we showed that FMNL2 participates in the assembly of actin stress fibers and induction of morphological changes of HTM cells. Actin stress fibers are bundles of contractile actomyosin that play an important role in mediating cellular adhesion and morphogenesis. The small GTPases RhoA, Rac1, and Cdc42 mainly regulate dynamics of actin assembly and disassembly. In general, RhoA is thought to

directly promote formation of actin stress fibers through its downstream effector, Rho-associated protein kinase (ROCK)[42]. Inhibitors of ROCK have been reported to disrupt actin stress fiber formation and modulate contractile activity of TM cells, thereby increasing aqueous outflow and lowering IOP[35,43]. Recently, an ophthalmic solution consisting of netarsudil 0.02% [Rhopressa®], a ROCK inhibitor has been approved in the United States for the reduction of elevated IOP in patients with POAG[44]. Another ophthalmic solution, ripasudil hydrochloride hydrate [Glanatec®], which is also a ROCK inhibitor, has been approved in Japan for the treatment of glaucoma and OHTN[45]. The small GTPase Cdc42 acts through its downstream mediators FMNL2 and FMNL3 to indirectly regulate the organization of the stress fibers by inducing formation of filopodia and promoting cell migration[33]. The function of FMNL2 in non-motile cells is not well defined. Our findings suggest a role of *FMNL2* in the assembly of actin stress fibers and for inducing morphological changes of TM cells, functions integral to the regulation of aqueous outflow and maintenance of normal IOP. The effect of FMNL2 on actin stress fiber formation is likely to be indirect via its influence on cell adhesion-based mechanisms that generate cellular tension and leading to RhoA activation[32,46].

We also identified a new region associated with POAG at *PDE7B* on chromosome 6. *PDE7B* encodes phosphodiesterase (PDE) 7B, a cAMP-specific PDE member of the cyclic nucleotide PDE family. The concentration of cAMP, a key second messenger of several biological processes is tightly regulated, by balancing its synthesis by adenylyl cyclases and its degradation by cAMP-specific PDEs. Pharmacologic inhibition or genetic inactivation of soluble adenylyl cyclase (sAC) in mice suppresses aqueous humor outflow, leading to elevated IOP[47]. Additionally, activation of sAC has been shown to elevate cAMP levels in RGCs, a critical step necessary for promoting RGC survival[48]. Thus, *PDE7B* through its effect on the regulation of cAMP levels may contribute to aqueous humor outflow or support RGC survival, functions that have important consequences in the pathogenesis of glaucoma.

The mechanisms that render RGCs susceptible to glaucomatous damage are largely unknown. Various insults including ischemia, excitotoxicity, inflammation, axonal injury, and glial activation have been proposed to contribute to RGC damage[49]. Degenerative programs both intrinsic and extrinsic to RGCs and their axons have been suggested to contribute to glaucomatous neurodegeneration[49,50]. Here we have identified a set of genes in POAG-associated loci whose expression is altered in the RGCs (e.g., *Ank*, *Cadm2*, and *Six6*) and the optic nerve head (e.g., *Cadm2* and *Cdkn2b*) in a mouse model of glaucoma. Some of these expression changes occur very early, even before their eyes show any signs of RGC loss, suggesting some of these genes may play a critical role in the pathogenesis of glaucomatous neurodegeneration. Future studies will determine if the identified genes are part of the pathways conferring susceptibility to glaucomatous neurodegeneration.

Our study also identified a new region on chromosome 12 associated with POAG, near *TMTC2*. The *TMTC2* locus has been previously reported to be associated with POAG in a Japanese population[51], with a different lead SNP (rs7961953), however, this association did not reach a genome-wide level of significance, possibly due to limited sample size (total of 827 Japanese POAG patients and 748 control subjects without glaucoma)[51]. More recently, a study of optic disc morphology identified genome-wide significant associations between variants near *TMTC2* and optic disc size and cup area[52]. Genetic variation in the *TMTC2* locus may contribute to glaucomatous neurodegeneration through its effect on optic nerve morphology.

Further, we identified *LMX1B* as a glaucoma susceptibility locus. *LMX1B* gene encodes LIM (Lin-1, Isl-1, and Mec-3)-homeodomain transcription factor 1 beta, and rare deleterious mutations in *LMX1B* cause NPS, with glaucoma being one of the manifestations of the syndrome[36,53]. Haploinsufficiency of *LMX1B* is generally accepted to cause NPS, but dominant negative effects are also possible[37,39]. Substantial phenotypic variability occurs among NPS patients with glaucoma occurring in about 30% of these patients[36,37]. In many of these patients, the glaucoma phenotype appears similar to open-angle glaucoma[37,38], with no obvious structural abnormalities in the anterior segment. In other patients, anterior segment developmental defects, including cataracts and abnormalities of the iris and cornea, are present, leading to the suggestion that *LMX1B* mutations contribute to the spectrum of glaucomas resulting from developmental defects[37]. Our findings demonstrate strong modifying effects of genetic background on ocular phenotypes induced by *Lmx1b* mutations, such that the mutations cause elevated IOP and glaucomatous nerve damage both in the presence and in the absence of obvious ocular developmental anomalies. This demonstration that *Lmx1b* mutations result in elevated IOP and glaucomatous nerve damage in eyes without developmental defects, supports the notion that *LMX1B* affects the risk of glaucoma in the absence of obvious developmental anomalies in human eyes.

Finally, our study reports differences in the prevalence of POAG across race/ethnicity groups, consistent with previous reports[54]. Within Hispanic/Latinos, East Asians, and African-Americans, the risk of POAG was associated with genetic ancestry. Specifically, a greater risk of POAG was associated with higher Native American (compared to European) ancestry among Hispanic/Latinos, higher northern East Asian (compared to southern East Asian) ancestry among East Asians, and higher African (compared to European) ancestry among African-Americans. After taking into account the effect of newly and previously identified SNPs, the ancestry association in African-Americans was attenuated, the association of ancestry in Hispanic/Latinos was nearly unchanged, and in East Asians the association with ancestry became slightly stronger. Thus, the identified POAG loci can explain some of the ancestry effects in African-Americans, but not Hispanic/Latinos and East Asians, suggesting that additional POAG risk loci that correlate with ancestry within East Asian and Hispanic populations remain to be discovered.

We recognize several potential limitations of our study. First, glaucoma diagnoses in UKB (our replication sample) were based on self-reported data, and the subtypes of glaucoma were unspecified, which may result in underestimates of the effects of individual SNPs due to phenotype misclassification. As a consequence, our replication analysis based on the UKB was likely underpowered relative to its sample size. However, in UKB, the proportion of glaucoma cases was relatively close to that reported for the global prevalence of glaucoma for population aged 40–80 years[1,2]. Second, the GERA non-Hispanic white participants were older on average than the GERA participants from the other race/ethnicity groups (Hispanic/Latino, East Asian, and African-American). This could explain the similar proportion of POAG cases in non-Hispanic whites and Hispanic/Latinos in our study, as advancing age has been shown to be a major risk factor for POAG[1,2]. Third, in our study, we note that the African-American subgroup has the smallest sample size compared to the other race/ethnicity groups, potentially limiting statistical power to detect some SNP associations. We note, however, that we did observe nominally significant associations with several previously reported and newly identified POAG risk loci in this group. Finally, although investigating all implicated genes in the current

study would be of great interest, we restricted our functional investigations to two genes (*FMNL2* and *LMX1B*) because they were implicated in the 95% credible set, were novel loci, replicated in an independent cohort, and resources were available for functional characterization. Further, the use of the transformed human NTM-5 cells as opposed to primary HTM cells to study FMNL2 function represents another limitation of the study.

In conclusion, we identified a total of 9 novel loci associated with the risk of POAG/glaucoma at a genome-wide significance level that replicated in an independent sample. Our multiethnic meta-analysis combining GERA and UKB identified an additional 24 novel loci that await validation. These loci explain some, but not all, of the observed effect of genetic ancestry on POAG susceptibility. Our expression analyses of RGCs and optic nerve head tissue suggest that several newly identified candidate genes may play a role in glaucomatous neurodegeneration. Finally, functional characterization of *FMNL2* and *LMX1B* points to a mechanistic effect on IOP levels, which supports their role as glaucoma risk genes.

## Methods

**GERA population.** We report a GWAS of glaucoma in 4 986 cases and 58 426 controls from the GERA cohort. The GERA cohort consists of 110 266 adults who consented to participate in the Research Program on Genes, Environment, and Health, established from the Kaiser Permanente Medical Care Plan, Northern California Region (KPNC)[55,56]. The Institutional Review Board of the Kaiser Foundation Research Institute has approved all study procedures.

**Case and control definition.** All GERA subjects included in this study had valid IOP measures, as previously described[31]. Briefly, non-numeric entries for IOP, extreme values (≤5 and >60 mm Hg), and measurements taken on a single eye were removed. Further, IOP measurements that were taken after initial prescription of IOP-lowering medications were excluded to avoid values influenced by treatment. Because IOP-lowering medications are almost always prescribed before IOP-lowering surgical interventions (i.e., laser trabeculoplasty, trabeculectomy, tube shunt procedures, etc.), we did not remove subjects who had these surgical interventions. Patients eligible for inclusion were identified from clinical diagnoses captured in the KPNC electronic health record (EHR) system. These clinical diagnoses were recorded in the EHR system as International Classification of Diseases, Ninth Revision (ICD-9) diagnosis codes. We defined glaucoma cases as having at least: (1) two diagnoses of POAG (ICD-9 codes 365.01, 365.1, 365.10, 365.11, and 365.15); or (2) two diagnoses of NTG (ICD-9 code 365.12); or (3) one diagnosis of POAG and one diagnosis of NTG. In all cases, at least one of the diagnoses was made by a Kaiser Permanente ophthalmologist. Further, the cases did not have any diagnosis of other subtypes of glaucoma (e.g., pseudoexfoliation, pigmentary, or PACG; ICD-9 codes 365.52, 365.13, and 365.2, respectively). After excluding subjects who have one or more diagnosis of any type of glaucoma (ICD-9 code, 365.xx), our control group included all the non-cases. Subjects who had no diagnosis of any type of glaucoma (any ICD-9 code 365.xx other than 365.04) but did have a diagnosis of OHTN (ICD-9 code 365.04), were included as controls. In total, 4986 POAG (including 743 NTG cases and 4 243 HTG cases) and 58 426 controls from GERA were included in this study.

**Genotyping and quality control and imputation.** GERA individuals' DNA samples were extracted using Oragene kits (DNA Genotek Inc., Ottawa, ON, Canada) at KPNC and genotyped at the Genomics Core Facility of UCSF. DNA samples were genotyped at over 665 000 genetic markers on four race/ethnicity-specific Affymetrix Axiom arrays (Affymetrix, Santa Clara, CA, USA) optimized for European, Latino, East Asian, and African-American individuals[57,58]. We performed genotype quality control (QC) procedures for the GERA samples on an array-wise basis[56]. Briefly, we included genetic markers with initial genotyping call rate ≥ 97%, genotype concordance rate > 0.75 across duplicate samples, and allele frequency difference ≤ 0.15 between females and males for autosomal markers. Approximately 94% of samples and over 98% of genetic markers assayed reached QC procedures. Moreover, genetic markers with genotype call rates < 90% were excluded, as well as genetic markers with a MAF < 1%.

We also performed imputation on an array-wise basis. Following the pre-phasing of genotypes with Shape-IT v2.r72719[59], we imputed genetic markers from the cosmopolitan 1000 Genomes Project reference panel (phase I integrated release; http://1000genomes.org) using IMPUTE2 v2.3.0[60].We used the information $r^2$ from IMPUTE2 as a QC parameter, which is an estimate of the correlation of the imputed genotype to the true genotype[61]. Genetic markers with an imputation $r^2 < 0.3$ were removed, and we restricted to markers that had a minor allele count ≥ 20.

**GWAS analysis and covariate adjustment.** We first analyzed each of the four self-reported race/ethnicity groups (non-Hispanic whites, Hispanic/Latinos, East Asians, and African-Americans) separately. We ran a logistic regression of POAG and each SNP using PLINK[62] v1.9 (www.cog-genomics.org/plink/1.9/) with the following covariates: age; sex; and ancestry principal components (PCs). We modeled data from each genetic marker using additive dosages to account for the uncertainty of imputation[63].

Eigenstrat[64] v4.2 was used to calculate the PCs on each of the 4 race/ethnicity groups, and subjects were included for analyses in their self-reported group, as previously described[55]. We included as covariates the top 10 ancestry PCs for the non-Hispanic whites, whereas we included the top 6 ancestry PCs for the 3 other race/ethnicity groups. To adjust for genetic ancestry, we also included the percentage of Ashkenazi (ASHK) ancestry as a covariate for the non-Hispanic white race/ethnicity group analysis[55]. The ASHK proportion was extracted from the initial European PC analysis, where individuals of European and ASHK ancestries were run together to produce eigenvectors. The clusters resulting from this were re-classified as 0.0, 0.5, 0.75, and 1.0 ASHK (by drawing grids in the PC1–PC2 space). A full description of the ancestry analyses is provided in Banda et al.[55].

**GERA meta-analysis of POAG.** We then undertook a GERA meta-analysis of POAG to combine the results of the four race/ethnicity groups using the R[65] (https://www.R-project.org) package "meta". We calculated fixed effects summary estimates for an additive model. We also estimated heterogeneity index, $I^2$ (0–100%) as well as P-value for Cochrane's Q statistic among groups. For each locus, we defined the top SNP as the most significant SNP within a 2 Mb window, and novel loci were defined as those that were located over 1 Mb apart from any previously reported locus.

**Plots of POAG prevalence vs. genetic ancestry.** To visualize the glaucoma distribution by the ancestry PCs, we created a smoothed distribution of each individual $i$'s glaucoma phenotype using a radial kernel density estimate weighted on the distance to each other $j$th individual, $\sum j\phi(\{d(i,j)/\max_{i',j'}[d(i',j')] \times k)\})$, where $\phi(.)$ is the standard normal density distribution, $k$ is the smooth value (5 for non-Hispanic whites; and 15 for East Asians, Hispanic/Latinos, and African-Americans), and $d(i',j')$ is the Euclidean distance based on the first two PCs. Race/ethnicity and/or nationality subgroup labels were derived from GERA or the Human Genome Diversity Project for visual representation of different groups[55].

**Secondary and subgroup analyses in GERA.** As a secondary analysis, we also assessed the associations between the 12 POAG-associated loci identified in GERA and IOP. IOP values were entered by clinicians at each vision encounter and were captured in the EHRs[31]. To measure IOP in KPNC ophthalmology practices, the major standard equipment used is a Goldmann applanation tonometer (Haag-Streit, Bern, Switzerland), followed by a non-contact tonometer (Nidek TonoRef II), a Tono-Pen XL, an iCare rebound tonometer (Tiolat, Helsinki, Finland) and other equipment, such as pneumotonometers. We evaluated individual's mean IOP from both eyes for each visit, and the individual's median of the mean across all the visits was used for analysis. A linear regression of IOP and each SNP was performed using PLINK v1.9 with age, sex, and ancestry PCs as covariates.

For the NTG and HTG analyses, we divided the POAG group into those subjects who met criteria for NTG (NTG group) and those subjects who met criteria for POAG but not NTG (HTG group). For the 373 subjects who met criteria for both POAG and NTG, an ophthalmologist (R.M.) chart reviewed each subject to categorize them into either group.

Conditional association analyses were conducted to identify additional independent SNPs at each locus by including all the 12 lead SNPs identified in the GERA multiethnic meta-analysis as covariates in the regression model. We evaluated whether any additional SNPs within a 2 Mb window (±1.0 Mb with respect to the original top SNP) achieved genome-wide significance. Associations that replicate at a Bonferroni-corrected significance threshold of 0.05/500 = 0.0001 (corresponding to an estimate of ~500 independent SNPs per locus for 2 Mb interval surrounding each of our original signals)[66] are reported. Finally, we conducted an epistasis analysis of all pairs of top SNPs in the four GERA race/ethnicity groups. For this analysis, a Bonferroni-corrected significance threshold of $0.05/264 = 1.9 \times 10^{-4}$ (accounting for the number of SNP pairs tested (12 × 11)/2, and for the four race/ethnicity groups) was applied.

**Replication analysis of previously reported SNPs in GERA.** To assess whether the 19 previously described POAG-associated loci replicated in the GERA cohort, we tested 19 statistically independent top SNPs previously found to be associated at a genome-wide level of significance ($N = 17$) or showing suggestive evidence ($N = 2$) of association ($P < 10^{-6}$)[8–18]. We reported a nominal significance level of 0.05 and a more stringent multiple testing correction accounting for the number of variants tested (Bonferroni-corrected alpha level of 0.0026 (=0.05/19)).

**GWAS heritability estimates and variance explained.** SNP-based heritability estimates were estimated for POAG using the GCTA software[25]. GCTA software computes the phenotypic variance explained by all analyzed genetic markers in the

genome by restricted maximum likelihood reached using expectation maximization. The analysis was restricted to autosomal markers, and we applied a genetic relationship matrix cutoff of 0.025. For statistical power purposes, the SNP-based heritability estimates analysis was conducted in the the non-Hispanic white race/ethnicity group, which is the largest group of individuals in GERA.

**UK Biobank.** To test the five novel GERA genome-wide significant SNPs for replication, we evaluated associations in the multiethnic UKB[23], which consisted of 7 329 glaucoma (subtype unspecified) cases and 169 561 controls from five race/ethnicity groups (European, East Asian, South Asian, African British, and mixed ancestries) (Supplementary Table 3). Imputation to the Haplotype Reference Consortium (HRC) has been described (www.ukbiobank.ac.uk), and imputation at a few non-HRC sites (for replication) was done pre-phasing with Eagle[67] and imputing with Minimac3[68] with the 1000 Genomes Project Phase I. The glaucoma phenotype was assessed through a touchscreen self-report questionnaire completed at the Assessment Centre, via the question "Has a doctor told you that you have any of the following problems with your eyes?", and cases were defined as those reporting "glaucoma." The control group included the non-cases and excluded subjects who reported "Prefer not to answer" or "Do not know."

**In silico analyses.** To provide a list of candidate genes for follow-up functional evaluation, a Bayesian approach (CAVIARBF)[69] was used, wich is publicly available at https://bitbucket.org/Wenan/caviarbf. This approach has been successfully used in previous studies[66,70]. For each of the associated signals, we computed each variant's capacity to explain the identified signal within a 2 Mb window (±1.0 Mb with respect to the original top variant) and derived the smallest set of variants that included the causal variant with 95% probability (95% credible set). A total of 1 098 variants in 28 annotated genes were included in these 24 credible sets (Supplementary Data 1).

We evaluated expression of the genes that contained associated 95% credible set variants in human ocular tissues using two publicly available databases: the OTDB[26]; and EyeSAGE[27,28] publicly available at https://genome.uiowa.edu/otdb/ and http://neibank.nei.nih.gov/EyeSAGE/index.shtml, respectively. The OTDB consists of gene expression data for 10 eye tissues from 20 normal human donors, and the gene expression is described as Affymetrix Probe Logarithmic Intensity Error (PLIER) normalized value[26].

**Gene expression in DBA/2J mice ocular tissues.** RNA-seq was performed on RGCs isolated from the retina of either 9-month-old D2 mice or a genetically matched control strain (substrain of D2 mice, D2-*Gpnmb*+ that do not develop high IOP/glaucoma)[29]. D2 and D2-*Gpnmb*+ mice used for the gene expression studies were from the The Jackson Laboratory, and experimental mice were all 9-month-old females (Gene Expression Omnibus accession number GSE90654). The D2 mouse is an age-dependent model of OHTN/glaucoma that closely mimics human pigmentary glaucoma[71]. In the D2 mouse, mutations in two alleles (*Gpnmb*[R150X] and *Tyrp1*[b]) drive progressive iris pigment dispersion and iris stromal atrophy that blocks and damages the ocular drainage tissue, leading to elevated IOP[72]. In our colony, D2 mice begin to exhibit high IOP around 6 months of age, and by 9 months of age (the age that we profiled) the majority of eyes have undergone periods of elevated IOP but are yet to develop any detectable RGC degeneration. Following this period of IOP elevation, the optic nerve progressively degenerates from 10 months onwards and degeneration is almost complete by 12 months of age (>70% of eyes have severe glaucoma)[71]. We chose the 9-month time point for RNA-sequencing analysis, as the majority of D2 eyes at this time point have intact axons despite experiencing high IOP, thereby providing us an avenue to capture very early disease relevant molecular changes prior to any noticeable axonal degeneration. We employed HC of transcriptomic datasets to generate unbiased unique clusters based on gene expression patterns within D2 samples. We identified four distinct groups; D2 group 1 being transcriptomically identical to the control strain (D2-*Gpnmb*+); and D2 groups 2–4 exhibiting increasing levels of dissimilarity compared to the control group at a transcriptomic level[29]. We compared gene expression profiles of each of the D2 groups with the control group. Out of the 28 genes that were implicated in the 95% credible set of variants, some did not have mouse homologs or available data (e.g., *LOC105389189*, *LOC145783*, *LMX1B*, and *ZNF280D*). We also examined the expression of *TMTC2* as our lead SNP that reached genome-wide level of significance in the GERA discovery cohort was near this gene. However, the true causal variant at this locus has <5% chance of being within *TMTC2*. We conducted adjustment for multiple testing using FDR (*q*), and genes were considered to be significantly differentially expressed at an FDR < 0.05.

We used the DATGAN software[30] to assess the expression of the genes that contained associated 95% credible set variants, in optic nerve head punches of D2 mouse model of glaucoma (http://glaucomadb.jax.org/glaucoma (doi:10.1186/1471-2164-12-429)). Briefly, the optic nerve head was dissected and profiled using Mouse 430 v2 arrays (Affymetrix) for 50 DBA/2J eyes and 10 DBA/2J-*Gpnmb*+ controls. HC was performed to identify groups of eyes undergoing molecular changes prior to morphological changes in glaucoma. Eyes were clustered into different stages using the expression profiles from the optic nerve head (Accession

number: GSE26299)[30]. Data represent the $\log_2$ fold change and *q*-value between D2 and D2.*Gpnmb*+ (control) samples. *q*-Values < 0.05 were considered significant.

**HTM cell culture and transfection.** Experiments were performed utilizing previously characterized HTM cell line NTM-5[73] (gift from Gulab Zode, University of North Texas Health Science, Fort Worth, Texas). HTM cells were cultured in Dulbecco's modified Eagle's medium and F12 containing 10% fetal bovine serum (FBS). For gene silencing, we used two separate siRNAs targeting different regions of *FMNL2*. siRNA1: SMARTpool ON-TARGETplus *FMNL2* siRNA (Dharmacon, #Cat L-031993-01). siRNA2: SMARTpool siGENOME *FMNL2* siRNA (Dharmacon #Cat M-031993-01). Non-targeting siRNA (Dharmacon #Cat D-001810-01) was used as a control siRNA. HTM cells were transfected either with *FMNL2* siRNA1, *FMNL2* siRNA2, or control siRNA. Briefly, $2 \times 10^5$ cells were added to wells containing siRNAs to achieve a final siRNA concentration of 100 nM. Reverse transfection was performed using Lipofectamine RNAiMAX (Invitrogen, Carlsbad, CA) according to the manufacturer's instructions and the cells were cultured for about 48 h to suppress expression of *FMNL2* before utilizing them for phenotypic characterization.

**Quantitative real-time polymerase chain reaction.** Total RNA was isolated from siRNA-transfected HTM cells using the Qiagen RNeasy with on-column DNase I treatment (Qiagen, Valencia, CA, USA) and reverse transcribed using the iScript cDNA Synthesis Kit (Bio-Rad, Hercules, CA, USA). Quantitative real-time polymerase chain reaction was performed on a Bio-Rad C1000 Thermal Cycler/CF96 Real-Time System using SsoAdvanced[TM] SYBR® Supermix (Bio-Rad). The primer sets used in our experiments are listed in Supplementary Table 12. Each reaction was run in technical duplicates and a minimum of four biological replicates used per group. The relative expression level of each gene was normalized to housekeeping genes (*ACTB* and *MAPK1*) and analyzed using the CFX manager software (Bio-Rad).

**Phalloidin staining and cell morphology and counting.** Following *FMNL2* gene silencing using siRNAs for 24 h, the HTM cells were serum-starved for another 24 h (no FBS). For the FBS-treated groups (+FBS), cell were exposed to media containing 10% FBS for 10 min or 1 h. HTM cells were then fixed with 3.7% paraformaldehyde and permeabilized using phosphate-buffered saline (PBS) containing 0.1% Triton X-100. To assess assembly of filamentous actin (F-actin), the HTM cells were then incubated with PBS containing 2 µM fluorescein isothiocyanate-labeled phalloidin for 30 min at 37 °C and finally mounted on a slide using VECTASHIELD mounting medium containing DAPI (Vector Laboratories Burlingame, CA, USA). Cells were visualized under fluorescence microscope (Zeiss LSM 700 laser scanning confocal microscope) to assess assembly of actin filament/stress fibers. To quantify changes in cellular morphology, the cells were categorized into three groups based on their shape, (1) rounded/deformed: cells that adhere poorly and appear as deformed or round; (2) intermediate: cells showing modest spreading; and (3) spread: cells that are well spread. Proportion of cells in each group was calculated and expressed as percentage. Three independent experiments were performed for each condition. At least 400 cells were counted for each experiment. The Pearson' chi-squared test was performed to evaluate the statistical significance among the groups. We used $2 \times 2$ and $2 \times 3$ contingency tables for −FBS and +FBS conditions, respectively.

**Mice and husbandry.** The *Lmx1b*[V265D] allele was backcrossed to the C57BL/6J (B6) and 129SvEvTac strain backgrounds for six or more generations. This allele is also known as the *Lmx1b*[Icst39]. Strains for backcrossing were obtained from The Jackson Laboratory (Bar Harbor, ME, USA). The *Lmx1b*[Q82X] allele is an independent and previously uncharacterized ENU-induced allele that was generated on a DBA/2J (D2) strain background that has a genetic sensitizer and is WT for the *Gpnmb*+ gene but by itself does not develop high IOP or glaucoma (D2.Cg-*Gpnmb*+*Cyp1b1*[tm1Gonz] strain background). Whole-genome sequencing (WGS) of the G1 with the *Lmx1b*[Q82X] allele allowed genetic mapping on this same strain background using custom markers developed from the pool of incidental mutations identified via WGS. The phenotype was mapped using over 500 meioses to a 6.7-Mb region where the *Lmx1b*[Q82X] allele is the only mutation predicted to affect gene function. The Animal Care and Use Committee of The Jackson Laboratory approved all experimental protocols. Mutant and control littermates were housed together in cages with Alpha-Dri bedding covered with polyester filters. Cages were maintained in an environment kept at 21 °C with a 14-h light:10-h dark cycle.

**Genotyping of the *Lmx1b* allele.** Genotype of the *Lmx1b*[V265D] and *Lmx1b*+ alleles was determined using an allele-specific PCR protocol. Genomic DNA was PCR amplified with forward primer for the V265D allele 5′-TCAGCGTGCGTGTGG TCCTGGA-3′, the forward primer for the WT allele 5′-GACATTGGCAGCAG AGACAGGCCGAGGCGTGCGTGTGGTCCATGT-3′, and the reverse primer 5′-ACACAAGCCTCTGCCTCCTT-3′. This same allele was previously named lcst[39]. Genomic DNA was PCR amplified using the following program: (1) 95 °C for 2 min; (2) 95 °C for 15 s; (3) 57 °C for 20 s; (4) 72 °C for 30 s; (5) repeat steps 2–4 35 times; and (6) 72 °C for 7 min. A volume of 5 µl of sample was run on a 3% agarose gel. The WT allele amplifies a 175-base pair fragment and the V265D allele

amplifies a 152-base pair fragment. Genotyping of the $Lmx1b^{Q82X}$ and $Lmx1b^+$ alleles was determined by direct Sanger sequencing of a specific PCR product. Genomic DNA was PCR amplified with forward primer 5′-CTTTGAGCC ATCGGAGCTG-3′ and reverse primer 5′-ATCTCCGACCGCTTCCTGAT-3′ using the following program: (1) 94 °C for 3 min; (2) 94 °C for 30 s; (3) 57 °C for 30 s; (4) 72 °C for 1 min; (5) repeat steps 2–4 35 times; and (6) 72 °C for 5 min. PCR products were purified and sequenced by the Genome Technologies service at The Jackson Laboratory.

**Clinical examination**. Detailed clinical evaluations were performed at 9 months of age using a slit lamp biomicroscope and photographed with a ×40 objective lens. Phenotypic evaluation included iris structure, pupillary abnormalities, generalized corneal haze, anterior and posterior synechia, corneal opacity, hyphaema, hypopyon, corneal pyogenic granuloma, vascularized scarred cornea, buphthalmos, cataracts, and deepening of the anterior chamber. Detailed examination of at least 30 eyes from each strain and genotype was performed. All cohorts included male and female mice.

**IOP measurement**. IOP was measured using the microneedle method as previously described in detail[74]. Briefly, mice were acclimatized to the procedure room and anesthetized via an intraperitoneal injection of a mixture of ketamine (99 mg/kg; Ketlar, Parke-Davis, Paramus, NJ) and xylazine (9 mg/kg; Rompun, Phoenix Pharmaceutical, St. Joseph, MO) prior to IOP assessment—a procedure that does not alter IOP in the experimental window[74]. All cohorts included male and female mice. The IOPs of B6 mice were assessed in parallel with experimental mice as a methodological control to ensure proper calibration and equipment function. The IOP values are highly variable in B6.$Lmx1b^{V265D/+}$ eyes. $Lmx1b$ mutants have abnormal corneas that are often stretched and sometimes perforate, and this perforation results in lower IOP values, which may explain the greater spread of IOP values in B6 $Lmx1b$ mutants. To evaluate the change in the range of IOP values in $Lmx1b$ mutant eyes across strains relative to controls, we used $k$-means clustering. We set $k = 3$ for each individual group. The cluster with the highest IOP values was taken from each group and a one-way analysis of variance was performed. Tukey's honest significance difference was used to compare the means between $Lmx1b$ mutant and WT mice within each strain.

**Ocular and optic nerve histological analysis**. Enucleated eyes were fixed for plastic sectioning as described in detail[75]. Serial sagittal sections were collected, stained with hematoxylin and eosin, and analyzed for pathologic alterations. Intracranial portions of optic nerves were dissected, processed, and analyzed as previously described and validated[71]. Briefly, we stained optic nerve cross sections with para-phenylenediamine (PPD) and then weinvestigated for glaucomatous damage. Because PPD stain differently stains the myelin sheaths and the axoplasm of healthy axons compared to those from sick or dying axons, we were able to detect axon damage and loss in a very sensitive way. The degree of nerve damage was assigned using a well-established protocol that has been extensively validated against axon counting[71,76]. Nerves with severe degeneration have >50% axon loss compared to controls, nerves with moderate moderate degeneration has on 30% average axon loss while nerves with no glaucoma are indistinguishable from controls. To determine whether optic nerve damage levels were dependent on $Lmx1b$ genotype, we compared mutant and control mice using Fisher's exact test, with $P < 0.01$ considered significant.

**MicronIV and image-guided optical coherence tomography**. Optical coherence tomography (OCT) paired with the MicronIV: Retinal Imaging Microscope (Phoenix Research Labs) was used to assess the retina and optic nerve head in vivo. Mice were subject to pupillary dilation with a drop of 1% solution of cyclopentolate hydrochloride ophthalmic solution (Akorn, Inc.) that was topically applied to the cornea. Mice were anesthetized as previously described (see methods, IOP measurement). Mice were secured on a rotating stage and imaged using OCT per the manufacturer's directions.

**Data availability**. Genotype data of GERA participants are available from the database of Genotypes and Phenotypes (dbGaP) under accession phs000674.v2.p2. This includes individuals who consented to having their data shared with dbGaP. The complete GERA data are available upon application to the KP Research Bank (https://researchbank.kaiserpermanente.org/). The meta-analysis GWAS summary statistics are available from the NHGRI-EBI GWAS Catalog (https://www.ebi.ac.uk/gwas/downloads/summary-statistics). The genotype data and the glaucoma phenotype of UKB participants are available upon request from (www.ukbiobank.ac.uk). The RNA-seq and microarray datasets utilized in the study were deposited previously on NCBI, accession number GSE90654 and GSE26299 respectively.

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

## Acknowledgements

We are grateful to the Kaiser Permanente Northern California members who have generously agreed to participate in the Kaiser Permanente Research Program on Genes, Environment, and Health. Support for participant enrollment, survey completion, and biospecimen collection for the RPGEH was provided by the Robert Wood Johnson Foundation, the Wayne and Gladys Valley Foundation, the Ellison Medical Foundation, and Kaiser Permanente Community Benefit Programs. Genotyping of the GERA cohort was funded by a grant from the National Institute on Aging, National Institute of Mental Health, and National Institute of Health Common Fund (RC2 AG036607 to C.S. and N.R.). Data analyses were facilitated by National Eye Institute (NEI) grant R01 EY027004 (E.J.) and a Kaiser Permanente Community Benefit grant (E.J.). This work was also made possible in part by NIH-NEI EY002162—Core Grant for Vision Research, by the Research to Prevent Blindness Unrestricted Grant (UCSF, Ophthalmology). K.S.N. receives support from NEI grant EY022891, Research to Prevent Blindness William and Mary Greve Special Scholar Award, Marin Community Foundation-Kathlyn McPherson Masneri and Arno P. Masneri Fund, and That Man May See Inc. S.W.M.J. receives support from NEI grant EY011721 and he is the investigator of a Howard Hughes Medical Institute (HHMI) grant. The funders had no role in study design, data collection and analysis, decision to publish, or preparation of the manuscript.

## Author contributions

H.C., K.S.N., and E.J. conceived and designed the study. T.J.H., M.N.K., N.R., C.S., and E.J. were involved in the genotyping and quality control. T.J.H. performed the imputation analyses. K.K.T., in collaboration with R.M., extracted phenotype data from EHRs. K.K.T. and J.Y. performed statistical analyses. Y.B. performed the ancestry principal components analyses. H.C. and J.Y. performed in silico analyses. S.P. and K.S.N. performed the functional characterization of *FMNL2*. S.C.K., N.G.T., K.S.N., and S.W.M.J. generated and characterized *Lmx1b* mutant mouse strains. P.A.W., N.G.T., and S.W.M.J. carried out the gene expression analysis in mouse glaucoma models. S.W.M.J. conceived and oversaw the mouse experiments. H.C., T.J.H., C.S., R.M., N.R., S.W.M.J., S.P., K.S.N., and E.J. interpreted the results of analyses. H.C., T.J.H., P.A.W, N.G.T., R.M.,

N.R., S.W.M.J., K.S.N., and E.J. contributed to the drafting and critical review of the manuscript.

## Additional information

**Competing interests:** The authors declare no competing interests.

