## [Peer Review File · Nature Communications]

Reviewer #1 (Remarks to the Author):

The novel and exciting study of Choquet and colleagues involved a multiethnic association study in the GERA cohort to identify 24 risk alleles for primary open-angle glaucoma (POAG), including 14 novel loci. 9 of the novel risk alleles were independently replicated in the UK Biobank. The authors prioritized these risk alleles and examined ocular tissue gene expression as well as differential expression of associated genes in RGCs and optic nerve heads of two mouse models of glaucoma. In addition, functional analyses were conducted on two of these risk genes FMNL2 and LMXL1. siRNA silencing of FMNL2 in a cultured TM cell line caused defective actin cytoskeleton formation, while two independent mutations in Lmx1b elevated IOP and optic nerve damage in mice. This scientifically rigorous and comprehensive study provides important new information about genetic risk factors responsible for POAG, the leading cause of irreversible vision loss and the leading neurodegenerative disease.

There are a number of issues that the authors should address to further improve their manuscript:

- (1) Please address the potential problems associated with unspecified glaucoma subtypes in the UKB cohort. It appears that glaucoma was self-reported (lines 501-502), which makes the glaucoma status much less reliable. Based on previous prevalence studies, what is the expected % of the individuals with glaucoma in the UK population that have POAG?
- (2) Line 71: Please specify that the subtype of glaucoma was not specified in this UKB cohort
- (3) Lines 92-93: Please provide potential explanations for either the higher prevalence of POAG in non-Hispanic whites (7.4%) or the lack of higher prevalence of POAG in the Hispanic/Latino subjects found in your study.
- (4) The use of the transformed human NTM5 cell line to evaluate the effect of LMNL2 knockdown is less than optimal. The NTM cells are morphologically different than primary HTM cells: they are smaller with less cytoplasm and fewer cytoskeletal elements, rounder, and not contact inhibited (constantly proliferating). Although NTM5 cells are generally easier to transfect, numerous groups have very successfully knocked down gene expression using siRNA in primary HTM cells. The results of LMNL2 knockdown should be even more compelling in primary HTM cells.
- (5) Figure 2: Which time point (10 minutes or 1 hr) was used for data presented in Figure 2C?
- (6) Lines 157-163: How was the IOP used for this data analysis determined? Was this the sum of all recorded IOPs? The highest recorded IOP? The authors need to discuss problems associated with evaluating IOP in therapeutically treated POAG patients, since this will vary considerably based on stage of disease. How many patients were treated with IOP lowering drops? How many patients had IOP lowering surgical interventions?

- (7) Lines 187-189: Please provide a potential explanation for lack of association for these two previously reported Asian POAG SNPs in the GERA meta-analysis.
- (8) Lines 242-243 and Figure 3B: The *Lmx1b* mutations only very modestly elevated mouse IOP by an average of 1-2 mmHg. Please discuss why this very modest IOP elevation caused significant optic nerve damage (Figure 3D). It is quite possible (although dismissed by the authors) that sedated daytime measurements of IOP are underestimating the true IOP elevations in these mutant mice.
- (9) Lines 650-653: What is the evidence for atrophy of the iris/ciliary body in the B6.*Lmx1b*^{V265D/+} mice as a potential explanation for the highly variable IOPs and the IOP “crashing”? Please use a more scientifically sound description instead of “crashing”.
- (10) Lines 887-889: The “glaucoma phenotypes for the UKB participants” could not be evaluated due to “off-line planned upgrade works”. Also, the authors stated that the glaucoma status of this UKB cohort was self-reported (Lines 501-502).

Reviewer #2 (Remarks to the Author):

In this elegant and multi-dimensional investigation, the authors performed a multi-ethnic GWAS of POAG and they identified novel loci, some of which were confirmed in a second population. To extend these findings further, they then used murine studies to delve into the plausible role that the gene products may play in the pathophysiology of glaucoma. The investigation is logical and very well presented. I have only a few comments that require clarification:

* It is not clear why *LMX1B* was excluded from Supplementary Figure 5. What is the significance of its absence?

* It is not clear why additional studies were performed on *FMNL2* and *LMX1B*, while the other loci were excluded? Please expand on the reasoning for this limited selection.

Reviewer #3 (Remarks to the Author):

The authors employ a large, ethnically diverse GERA cohort, to discover 5 novel genome-wide significant POAG loci, of which 3 (*FMNL2*, *PDE7B*, and near *TMTC2*) replicated in the UK Biobank cohort after adjustment for multiple comparisons. They also investigated 9 novel glaucoma associated loci from UKB in GERA, and 6 of the novel loci replicated at Bonferroni significance (near

IKZF2, CADM2, near DGKG, ANKH, EXOC2, and LMX1B). Furthermore, a multiethnic meta analysis combining GERA and UKB identified an additional 24 novel loci that await validation in an external replication cohort. The authors went on to perform functional studies to establish a role for FMNL2 and LMX1 in the pathogenesis of primary open angle glaucoma. Overall the GERA and UK Biobank cohorts are emerging as powerful resources to provide insight into the pathogenesis of glaucoma.

In the introduction the authors state that “many of the reported loci have not yet been validated in an independent study, nor have their roles been investigated in functional studies.” This is not at all true. Actually most of the common loci for POAG discovered in gwas have been replicated, otherwise they would not have been published - the standard in the field is to confirm loci discovered via agnostic gene search. Furthermore while functional studies of many POAG variants are lacking some interesting functional work has been done for CDKN2B-AS, SIX6, and CAV1/2. For example there is the nice study by Gao and Jakobs entitled,

Mice Homozygous for a Deletion in the Glaucoma Susceptibility Locus INK4 Show Increased Vulnerability of Retinal Ganglion Cells to Elevated Intraocular Pressure. , Am J Pathol. 2016 Apr;186(4):985-1005. doi: 10.1016/j.ajpath.2015.11.026. Epub 2016 Feb 13.

The authors should modify their assertions on this matter.

The supplementary material should have a table of contents

The rationale for the high genomic inflation factor is a little concerning. Upon review of the supplemental Figure 1, the high genomic inflation factor is driven by non-Hispanic whites. This raises concerns about cryptic relatedness in this subpopulation. The authors need to address how they minimized cryptic relatedness in GERA.

In the main results section, the authors should comment regarding the extent to which the new loci replicate in Hispanics, Asians and African people.

In line 159, the authors mention that they assess the lead 12 glaucoma SNPs in relation to IOP. They should clarify that this is IOP measured in GERA not UKB. In the methods section the authors should clarify what they mean by “valid IOP measurements”. I assume this also means IOP as measured in cases and controls. How was IOP adjusted for cases that were on glaucoma treatment?

On line 243, the authors state, “ IOP progressively crashes in some of the

244 B6.Lmx1bV265D/+ mice.” This is vague jargon. Please explain exactly what this statement means.

Reference 22 is incomplete.

The following statement is not up to date: “In fact, clinical trials to test the efficacy of ROCK inhibitors are underway to lower IOP in patients with glaucoma.” The FDA in the US approved Netarsudil and Ripasudil is approved for clinical use in Japan.

Overall the results are novel and interesting. Despite some methodology concerns about how POAG was diagnosed the findings are convincing. The functional studies do not address the discovered variants and this is a weakness.

Reviewer #4 (Remarks to the Author):

This manuscript describes a large body of work to identify genetic risk loci for primary open angle glaucoma (POAG). The authors begin with a GWAS meta-analysis of 4 ethnic groups from the GERA cohort, which is dominated by non-Hispanic whites, but also includes Hispanic/Latino, East Asian and African Americans. Replication of the findings are undertaken in the publicly available UK Biobank data. A full GWAS of UK biobank is also presented, with subsequent replication in the GERA cohort. Finally, a meta-analysis of GERA and UKB is presented, along with secondary analyses for IOP and conditional association. The authors have also explored the biology of the associated loci, evaluating actin stress fibre formation in a cell culture model, examining the glaucoma phenotype in a mouse mutant, and reporting the ocular tissue expression patterns in mouse and human of genes at the associated loci.

Overall, the authors report 9 novel loci for POAG with replication as well as 24 novel loci from the final meta-analysis, representing a substantial contribution to POAG genetics. Some additional information would clarify some sections.

Comments for the authors

1. The analysis of ancestry vs prevalence is really interesting, but I spent a long time trying to interpret Figure 1 due to scant methodological information on the generation of these plots. Please describe clearly how the prevalence is calculated for these plots. My assumption is that it was calculated per 'bin' of PC1 or PC2, but it is not at all clear
2. Were outliers removed based on PCA in any of the cohorts? IF not, why not? The PCA plot for Hispanic/Latino appears to show individuals with strong African heritage. Were these individuals left in the Hispanic cohort or moved to the African American cohort based on the observed genetic ancestry?

3. Table 1: shows NTG cases, which seems to relate to the analyses in sup Table 5, but what are the “OHTN controls” and where do they come into subsequent analyses? There is no dichotomised analysis of OHTN presented and the number seems very low if the definition is of IOP in the normal range.

4. Supp Table 1: “GERA” covariate association with age, sex and the ancestry PCs”. Please confirm (in the table title) that this table is showing the association of each covariate (age, sex, ancestry PCs) with POAG in the GERA cohort. Also, please define how “Ashkenazi” is defined and where that data came from for the non-hispanic whites

5. The order of ethnicities changes between Figure 1 and Supp Figure 1. Please use the same order of sub-groups in all Tables and Figures to make it easier for the reader to follow

6. Supp Figure 3: Please indicate the source data of the linkage disequilibrium and recombination rate on the LocusZoom plots.

7. Table 2 (and other tables with Odds Ratios): Please provide the confidence intervals on the ORs

8. Figure 2 and Supp Figure 4:

a. The differences in phalloidin staining between control and siRNA in the presence of FBS are not particularly convincing in Figure 2B. Supp Figure 4 more clearly shows reduced labelling in some cells. Note there is a typo in the figure legend (refers to Figure 1B instead of 2B). Please consider incorporating the Supp Figure images into the main figure to more clearly show the claimed result of altered actin stress filaments. The results for one of the two siRNAs could be moved to supplementary instead.

b. Also, which siRNA is used for the data presented in 2C? Finally, the conclusion of this paragraph (page 7, line 144) claims ‘reduced formation of actin stress fibres’. This may be the case, but the fibres have not been quantitated and the images don’t necessarily support this claim, depending which cell you look at. Please discuss this in more detail and alter the conclusion appropriately.

c. The quantitation of the morphology seems more robust, although it is not clear how the statistics were handled from the methods. Is it a 2x3 chi-square test for the three morphology groups, or a 2x2, with a collapsing of groups?

d. Can “intermediate” and “well-spread” cells be indicated on the figure, to show the difference between these two morphologies?

e. Methods for this section, page 26 line 587: Were the cells serum starved immediately following transfection, or after 48 of growth? If the latter, was knockdown maintained through the 72 hours post-transfection?

9. Supp Table 5: How were the lead SNPs classified into high and normal IOP in this table and why? Is it based on association with IOP? Or NTG? Several loci in the 'high IOP' half of the table have negative associations with IOP, indicating the minor allele is associated with decreased IOP. While this means the other allele is associated with increased pressure, it is counter-intuitive with the labelling in this table and should be clarified. It would be easier to follow if the table columns matched the order of the description in the text on page 8 (i.e. IOP, NTG, HTG). The methods state that a GWAS was conducted for IOP, but only specific loci taken from the POAG GWAS are presented. Either present the full results, or modify the methods. Also, which statistic was used for the IOP analysis and were any covariates included?

10. Supp Table 7: Why is the NCKAP5 locus not reported in the meta-analysis, even though the SNP is present in all 4 populations?

11. Does the lead SNP at the FMNL2, or one in strong LD with it, influence the expression of the FMNL2 gene, or any other nearby gene? These data can be accessed from the GTex project and could be informative for putative functional SNPs at all replicated loci, albeit, with the caveat of the specific tissues available in GTex.

12. Discussion page 15 line 349-352: please indicate here which genes specifically you are referring to, given that the data is in a supplementary figure. Are there any known pathway connections between these genes?

13. The overall structure of the paper is quite confusing. The methods are results are ordered differently, making it very confusing to move between sections. The discussion is different again. Please at least make the methods and results consistent. The results may be easier to follow if they present all the GWAS data, then the gene expression data for discovered loci, then the specific functional analyses of FMNL2 and Lmx1b.

We thank the reviewers for their helpful comments, and their enthusiastic support, including that “this scientifically rigorous and comprehensive study provides important new information about genetic risk factors responsible for POAG”, “exciting study”, “elegant and multi-dimensional investigation”, “the investigation is logical and very well presented” and that “the findings are convincing” and represent “a substantial contribution to POAG genetics”. The reviewers made excellent suggestions to improve our manuscript, including: 1) to comment regarding the extent to which the new loci replicate in Hispanics, Asians, and African Americans; 2) to assess whether the lead SNP at *FMNL2* influences the expression of the *FMNL2* gene, or any other nearby gene; and 3) to make consistent the structure of the paper by ordering similarly the methods and results sub-sections. We have followed these suggestions, and made changes to the manuscript to address these points. Below, we provide detailed responses addressing the individual comments of the reviewers.

Reviewers' comments:

Reviewer #1 (Remarks to the Author):

The novel and exciting study of Choquet and colleagues involved a multiethnic association study in the GERA cohort to identify 24 risk alleles for primary open-angle glaucoma (POAG), including 14 novel loci. 9 of the novel risk alleles were independently replicated in the UK Biobank. The authors prioritized these risk alleles and examined ocular tissue gene expression as well as differential expression of associated genes in RGCs and optic nerve heads of two mouse models of glaucoma. In addition, functional analyses were conducted on two of these risk genes *FMNL2* and *LMXL1*. siRNA silencing of *FMNL2* in a cultured TM cell line caused defective actin cytoskeleton formation, while two independent mutations in *Lmx1b* elevated IOP and optic nerve damage in mice. This scientifically rigorous and comprehensive study provides important new information about genetic risk factors responsible for POAG, the leading cause of irreversible vision loss and the leading neurodegenerative disease.

Thank you to the reviewer for the positive feedback.

There are a number of issues that the authors should address to further improve their manuscript:

- 1. Please address the potential problems associated with unspecified glaucoma subtypes in the UKB cohort. It appears that glaucoma was self-reported (lines 501-502), which makes the glaucoma status much less reliable. Based on previous prevalence studies, what is the expected % of the individuals with glaucoma in the UK population that have POAG?**

In the UK Biobank, 7,329 individuals self-reported having glaucoma, which corresponds to a sample proportion of 4.1%. This proportion is slightly higher in comparison to a prevalence of 3.5% for population aged 40 or older (Tham *et al.* Ophthalmology 2014; Jonas *et al.* Lancet 2017). POAG makes up a majority of glaucoma, so we would expect a somewhat higher proportion in the UKB cohort. Also, we would expect self-report of glaucoma to be less accurate than our GERA clinically diagnoses POAG patients. The result of this type of phenotype misclassification would be that our analyses may underestimate the effects of individual SNPs.

We have added a sentence in the Discussion to reflect this limitation, as below:

“We recognize several potential limitations of our study. First, glaucoma diagnoses in UK Biobank (our replication sample) were based on self-reported data, and the subtypes of glaucoma were unspecified, which may result in underestimates of the effects of individual SNPs due to phenotype misclassification. As a consequence, our replication analysis based on the UK Biobank was likely underpowered relative to its sample size. However, in UK Biobank, the proportion of glaucoma cases was relatively close to that

reported for the global prevalence of glaucoma for population aged 40-80 years (Tham *et al.* Ophthalmology 2014; Jonas et al. Lancet 2017).”

As POAG is the most common type of glaucoma accounting for three-quarters of all glaucoma cases (Quigley et al. Br J Ophthalmol 2006), we expect that approximately 75% of the individuals with glaucoma in UKB have POAG.

2. Line 71: Please specify that the subtype of glaucoma was not specified in this UKB cohort

We have added this important information in the last paragraph of the Introduction, as below:

“We tested novel loci discovered in the current study in an independent external replication cohort: the multiethnic UK Biobank (UKB) which includes 7,329 glaucoma cases (subtype unspecified) and 169,561 controls.”

3. Lines 92-93: Please provide potential explanations for either the higher prevalence of POAG in non-Hispanic whites (7.4%) or the lack of higher prevalence of POAG in the Hispanic/Latino subjects found in your study.

The non-Hispanic white GERA participants are older on average than the GERA members from the other race/ethnicity groups (Hispanic/Latino, East Asian, and African-American). In our study, the average age at last vision exam was slightly higher in non-Hispanic white cases than in Hispanic/Latino cases (80.5 ± 9.0 vs. 75.5 ± 10.1 , respectively). Consistent with a previous study (Quigley, H. A. et al. Arch Ophthalmol 2001 - PMID: 11735794), we found that the proportion that are POAG cases in Hispanic/Latinos (7.9%) was between the proportion that are POAG cases in non-Hispanic whites (7.4%) and the proportion that are POAG cases in African Americans (16.1%). Further, it has been shown that in Hispanic/Latinos, the prevalence of POAG increased more quickly with increasing age than in other race/ethnicity groups¹. This could explain the higher prevalence of POAG in our study, in particular in Hispanic/Latinos and non-Hispanic whites as advancing age has been shown to be a major risk factor for POAG.

We have added information on age and sex for cases and controls for each race/ethnicity group in Table 1.

We have also added a sentence in the discussion to reflect this limitation as below:

“We recognize several potential limitations of our study ... Second, the GERA non-Hispanic white participants were older on average than the GERA participants from the other race/ethnicity groups (Hispanic/Latino, East Asian, and African-American). This could explain the similar proportion of POAG cases in non-Hispanic whites and Hispanic/Latinos in our study, as advancing age has been shown to be a major risk factor for POAG.”

4. The use of the transformed human NTM5 cell line to evaluate the effect of LMNL2 knockdown is less than optimal. The NTM cells are morphologically different than primary HTM cells: they are smaller with less cytoplasm and fewer cytoskeletal elements, rounder, and not contact inhibited (constantly proliferating). Although NTM5 cells are generally easier to transfect, numerous groups have very successfully knocked down gene expression using siRNA in primary HTM cells. The results of LMNL2 knockdown should be even more compelling in primary HTM cells.

We agree with the reviewer that using primary HTM cells would have been very useful. However, despite our attempts we were not able to induce appreciable downregulation of Fmnl2 transcripts in primary HTM cells due to technical difficulties. We feel our experiments using NTM5 are a good starting point and suggest an important role of Fmnl2 in supporting contractile properties of TM cell.

5. Figure 2: Which time point (10 minutes or 1 hr) was used for data presented in Figure 2C?

We have revised the figure legend to clarify that we present data for both 10 minutes (FBS 10 min) and 1 hour (FBS 1hr) in Figure 3C (originally Figure 2C).

6. Lines 157-163: How was the IOP used for this data analysis determined? Was this the sum of all recorded IOPs? The highest recorded IOP? The authors need to discuss problems associated with evaluating IOP in therapeutically treated POAG patients, since this will vary considerably based on stage of disease. How many patients were treated with IOP lowering drops? How many patients had IOP lowering surgical interventions?

For this secondary analysis, IOP was determined in GERA cohort by: 1) assessing the individual's mean IOP from both eyes for each visit, and then 2) assessing the individual's median of these mean values across all the visits. To exclude values influenced by glaucoma treatment, we removed 167,293 IOP measurements from 4,786 POAG patients that were taken after the initial prescription of IOP lowering medications. Further, as IOP lowering medications are always prescribed before IOP lowering surgical interventions, the IOP measurements included in our analysis are values taken prior to any surgical interventions.

We have now provided more details in the Results and Methods sections, as below:

In the Results:

“Secondary and sub-group analyses

To further investigate whether the POAG susceptibility loci identified in this study influence glaucoma susceptibility through their effect on IOP or independently of IOP, we conducted three additional analyses. First, we examined the association of lead SNPs at these loci with IOP, which was assessed in GERA cohort (see Methods).”

In the Methods:

“Case and control definition

All GERA subjects included in this study had valid IOP measures as previously described (Choquet H, et al. Nat Commun. 2017). Briefly, non-numeric entries for IOP, extreme values (≤ 5 and >60 mmHg), and measurements taken on a single eye were removed. Further, IOP measurements that were taken after initial prescription of IOP lowering medications were excluded to avoid values influenced by treatment”.

“GWAS analysis and covariate adjustment

As a secondary analysis, we also conducted a GWAS of IOP. IOP measurements as entered by clinicians at each vision encounter were captured in the electronic health records as smart variables. The main standard equipment for measuring IOP in KPNC ophthalmology practices is a Goldmann applanation tonometer (Haag-Streit, Bern, Switzerland), followed by a non-contact tonometer (Nidek TonoRef II), a Tono-Pen XL, an iCare rebound tonometer (Tiolat, Helsinki, Finland) and other equipment, including pneumotonometers. Individual's mean IOP from both eyes for each visit was assessed, and the individual's median of the mean across all the visits was used for analysis.”

7. Lines 187-189: Please provide a potential explanation for lack of association for these two previously reported Asian POAG SNPs in the GERA meta-analysis.

In the GERA sample, *NCKAP5* rs7588567 had a moderate imputation quality score r^2 in non-Hispanic whites ($r^2=0.64$), as well as in Hispanic/Latinos ($r^2=0.63$) and in African-Americans ($r^2=0.61$). For this reason, rs7588567 was excluded from the meta-analysis across the 4 race/ethnicity groups. In contrast, the imputation quality score r^2 for rs7588567 was excellent in East Asians ($r^2=0.99$). However, this SNP

was not associated with POAG in East Asians (OR=1.03, $P=0.65$), and showed inconsistent direction of effect in comparison to the effect reported in the initial study (Osman et al. HMG 2012). In the discovery cohort (Osman et al. HMG 2012), the association between rs7588567 and POAG did not achieve genome-wide level of significance ($P=4.79\times 10^{-8}$) and was not validated in the replication sample ($P=0.07$). Further, in the initial study, authors emphasized the need of further validation for this single-marker association which might be due to the “high recombination rates in this locus”.

PMM2 rs3785176 which was reported with suggestive evidence of association with POAG ($P=3.18\times 10^{-6}$) in a previous GWAS conducted in Chinese populations (Chen et al. NG 2014), was not associated with POAG in the current study.

The absence of replication for these two SNPs in our study is unlikely to be due to a lack of statistical power. We have assessed the statistical power using QUANTO software, with the following parameters: total number of cases and controls in GERA full sample (or in GERA East Asian race/ethnicity group), the prevalence of 3% for POAG in the general population, the effect size (ORs) from the previous studies and an α -level of 0.05. We had greater than 80% power to detect the previously reported OR for both SNPs rs7588567 and rs3785176 in both GERA full sample and GERA East Asian race/ethnicity group. Our findings clearly exclude any association of rs7588567 and rs3785176 with POAG in our sample.

We have added some text in the Results section to reflect this point, as below:

“In contrast, two SNPs (at *NCKAP5* and *PMM2*), which were reported with suggestive evidence of association with POAG in previous studies of Asian individuals, were associated with POAG in neither the current GERA multiethnic meta-analysis ($P > 0.05$), nor in the East Asian race/ethnicity group. The absence of replication in our study is unlikely to be due to a lack of statistical power, as we have estimated that our study of 4,986 cases and 58,426 controls (full GERA sample) or 441 cases and 4,034 controls (GERA East Asian race/ethnicity group) had more than 80% power to detect the previously reported effects for both SNPs rs7588567 and rs3785176. We note that in our GERA sample, *NCKAP5* rs7588567 had a moderate imputation quality score r^2 in non-Hispanic whites ($r^2=0.64$), as well as in Hispanic/Latinos ($r^2=0.63$) and in African-Americans ($r^2=0.61$). For this reason, rs7588567 was excluded from the meta-analysis across the 4 race/ethnicity groups.”

We have also added a note below the Supplementary Table 7 in the Supplementary Information file:

“*NCKAP5* rs7588567 had a moderate imputation quality score r^2 in non-Hispanic whites ($r^2=0.64$), as well as in Hispanic/Latinos ($r^2=0.63$) and in African-Americans ($r^2=0.61$). For this reason, rs7588567 was excluded from the meta-analysis across the 4 race/ethnicity groups.”

- 8. Lines 242-243 and Figure 3B: The *Lmx1b* mutations only very modestly elevated mouse IOP by an average of 1-2 mmHg. Please discuss why this very modest IOP elevation caused significant optic nerve damage (Figure 3D). It is quite possible (although dismissed by the authors) that sedated daytime measurements of IOP are underestimating the true IOP elevations in these mutant mice.**

This is a good point raised by the reviewer. The IOP values in individual eyes are far more important, especially given the spread of values. To clarify, we have now reformatted the presentation of the IOP data in Figure 4 (originally Figure 3). We used k-means clustering, a standard statistical approach for the unbiased identification of subgroups with differential responses in a group or data set. By color coding the IOP distributions of each genotype into their respective high, medium, and low IOP subgroups, it is easier to see the high IOP values.

We agree with the reviewer's comment about day time measurements, but this is the only practical way we can conduct the experiments. Importantly, it is not technically feasible to continuously measure mouse IOP, and their IOP can vary considerably with time of day, typically being the highest at night. This is also true for mice of strain DBA/2J where the highest IOP is at night, especially for the Q82X mutants (Savinova, 2001). Thus, it is quite likely we are underestimating the number of mice with elevated IOP and the magnitude of the elevation. Although sedation does provide a risk of lowering IOP, we have extensively validated our approach and anesthesia regimen so that we always measure IOP during an experimental window when no effect of anesthesia is detectable (Savinova, 2001).

9. Lines 650-653: What is the evidence for atrophy of the iris/ciliary body in the B6.Lmx1b^{V265D/+} mice as a potential explanation for the highly variable IOPs and the IOP “crashing”? Please use a more scientifically sound description instead of “crashing”.

Lmx1b mutants have abnormal corneas that are often stretched and sometimes perforate. This could potentially explain the highly variable IOPs, especially considering that B6 Lmx1b mutants have the most severely affected corneas as well as the lowest IOPs. Alternatively, and as often observed with mutant genes that elevate IOP, high IOP is only detected in a subset of mice at any time. The overall IOP distribution is often broadened and can include IOP values that are lower than normal. This is believed to be due to altered IOP homeostasis, including abnormal diurnal regulation, in response to genetic and physiologic changes (John, 1998 & Chang, 2001 & Smith, 2000). Alternatively and although we have not examined the ciliary body (which makes aqueous humor) in Lmx1b mutant mice, there is precedent that high IOP can induce ciliary body atrophy in genetically susceptible humans and mice, and ciliary body atrophy occurs in DBA/2J mice (John, 1998). Additionally, variability in IOP may result from stochastic variation in developmental consequences resulting from the Lmx1b mutation. As precedents, various mutant genes that affect development of ocular drainage structures also variably result in maldevelopment of the ciliary body (Smith, 2000 & Chang, 2001).

We recognize that in our original description the term “crashing” was inappropriate and we now provide a more scientifically sound description in the Results and Methods sections, as follows:

“The effect of Lmx1b mutations on glaucoma-related phenotypes depends on genetic background

Slit lamp-based clinical eye examination showed that the B6 mutant mice exhibit a severe developmental phenotype characterized by malformed eccentric pupils, irido-corneal strands, corneal haze and corneal scleralization (Figure 4.A.). The developmental phenotype on a 129 background was much milder, and mainly limited to mild pupillary abnormalities in about half of the mice. No major developmental abnormalities were detected in D2 mice with the *Lmx1b*^{Q82X} allele (focal corneal keratopathy is an unrelated strain characteristic of D2 mice that is frequently present in both wildtype and mutant mice). Further, with age, high IOP often results in a more open pupil configuration in D2.*Lmx1b*^{Q82X/+} mice. The B6.*Lmx1b*^{V265D/+} and D2.*Lmx1b*^{Q28X/+} mice were highly susceptible to developing glaucomatous nerve damage (Figure 4.B. and 4.C.), while 129.*Lmx1b*^{V265/+} mice rarely developed nerve damage (data not shown). Finally, Lmx1b mutations induced elevated IOP in all three strain backgrounds (Figure 4.D. and 4.E.). IOP is highly variable in the B6.*Lmx1b*^{V265D/+} which is possibly caused by a variety of reasons. One possible explanation for the spread of IOPs is due to Lmx1b mutants exhibiting abnormal corneas that are often stretched and sometimes perforate. This is a likely explanation, especially considering that B6 *Lmx1b* mutants have the most severely affected corneas as well as the lowest IOPs. Additionally, variability in IOP may result from stochastic variation in developmental consequences resulting from the *Lmx1b* mutation. As precedents, various mutant genes that affect development of ocular drainage structures also variably result in maldevelopment of the ciliary body (Smith, 2000 & Chang, 2001).”

“IOP measurement

IOP was measured using the microneedle method as previously described in detail⁹⁰. Briefly, mice were acclimatized to the procedure room and anesthetized via an intraperitoneal injection of a mixture of ketamine (99 mg/kg; Ketlar, Parke-Davis, Paramus, NJ) and xylazine (9 mg/kg; Rompun, Phoenix Pharmaceutical, St. Joseph, MO) prior to IOP assessment - a procedure that does not alter IOP in the experimental window. All cohorts included male and female mice. The IOPs of B6 mice were assessed in parallel with experimental mice as a methodological control to ensure proper calibration and equipment function. The IOP values are highly variable in B6.*Lmx1b*^{V265D/+} eyes. *Lmx1b* mutants have abnormal corneas that are often stretched and sometimes perforate, and this perforation results in lower IOP values, which may explain the greater spread of IOP values in B6 *Lmx1b* mutants. To evaluate the change in the range of IOP values in *Lmx1b* mutant eyes across strains relative to controls, we used k-means clustering. We set k=3 for each individual group. The cluster with the highest IOP values was taken from each group and a one-way analysis of variance (ANOVA) was performed. Tukey’s honest significance difference (HSD) was used to compare the means between *Lmx1b* mutant and WT mice within each strain.

10. Lines 887-889: The “glaucoma phenotypes for the UKB participants” could not be evaluated due to “off-line planned upgrade works”. Also, the authors stated that the glaucoma status of this UKB cohort was self-reported (Lines 501-502).

We did not foresee that the UKB website would be unavailable in the last few weeks. To avoid this issue going forward, we now provide the more general UKB URL in the Data Availability paragraph as follows:

“The genotype data and the glaucoma phenotype of UKB participants are available upon request from (www.ukbiobank.ac.uk).”

Reviewer #2 (Remarks to the Author):

In this elegant and multi-dimensional investigation, the authors performed a multi-ethnic GWAS of POAG and they identified novel loci, some of which were confirmed in a second population. To extend these findings further, they then used murine studies to delve into the plausible role that the gene products may play in the pathophysiology of glaucoma. The investigation is logical and very well presented.

We thank the reviewer for the supportive comments.

I have only a few comments that require clarification:

*** It is not clear why LMX1B was excluded from Supplementary Figure 5. What is the significance of its absence?**

Lmx1b was not included in this supplementary figure because Lmx1b mRNA was not detected in mouse retinal ganglion cells (count per million was zero).

We have added text in the Methods section to clarify this point, as below:

“Out of the 28 genes that were implicated in the 95% credible set of variants, some did not have mouse homologs or available data (e.g. *LOC105378189*, *LOC145783*, *LMX1B*, and *ZNF280D*).”

*** It is not clear why additional studies were performed on FMNL2 and LMX1B, while the other loci were excluded? Please expand on the reasoning for this limited selection.**

While we would like to investigate all implicated genes in our study, we had limited resources and time in which to do so for the current paper. We decided to investigate these two genes, because they were implicated in the 95% credible set, were novel loci, replicated in an independent cohort, and they had resources available for functional characterization.

We have added a sentence in the Discussion to reflect this limitation, as below:

“Finally, although investigating all implicated genes in the current study would be of great interest, we restricted our functional investigations to two genes (*FMNL2* and *LMX1B*) because they were implicated in the 95% credible set, were novel loci, replicated in an independent cohort, and resources were available for functional characterization.”

Reviewer #3 (Remarks to the Author):

The authors employ a large, ethnically diverse GERA cohort, to discover 5 novel genome-wide significant POAG loci, of which 3 (FMNL2, PDE7B, and near TMTC2) replicated in the UK Biobank cohort after adjustment for multiple comparisons. They also investigated 9 novel glaucoma associated loci from UKB in GERA, and 6 of the novel loci replicated at Bonferroni significance (near IKZF2, CADM2, near DGKG, ANKH, EXOC2, and LMX1B). Furthermore, a multiethnic meta analysis combining GERA and UKB identified an additional 24 novel loci that await validation in an external replication cohort. The authors went on to perform functional studies to establish a role for FMNL2 and LMX1 in the pathogenesis of primary open angle glaucoma. Overall the GERA and UK Biobank cohorts are emerging as powerful resources to provide insight into the pathogenesis of glaucoma.

We are grateful to this reviewer for his/her careful attention to our paper.

1. In the introduction the authors state that “many of the reported loci have not yet been validated in an independent study, nor have their roles been investigated in functional studies.” This is not at all true. Actually most of the common loci for POAG discovered in gwas have been replicated, otherwise they would not have been published - the standard in the field is to confirm loci discovered via agnostic gene search. Furthermore while functional studies of many POAG variants are lacking some interesting functional work has been done for CDKN2B-AS, SIX6, and CAV1/2. For example there is the nice study by Gao and Jakobs entitled, Mice Homozygous for a Deletion in the Glaucoma Susceptibility Locus INK4 Show Increased Vulnerability of Retinal Ganglion Cells to Elevated Intraocular Pressure. , Am J Pathol. 2016 Apr;186(4):985-1005. doi: 10.1016/j.ajpath.2015.11.026. Epub 2016 Feb 13. The authors should modify their assertions on this matter.

We agree with the reviewer that this introduction statement in the original manuscript was not accurate. We have modified this statement regarding previous genetic studies in the Introduction section, as following:

“Genome-wide association studies (GWAS) have reported 17 loci associated with POAG at genome-wide significance, and an additional 2 loci at a suggestive level of significance ($P < 10^{-6}$)⁹⁻¹⁹. Together these loci explain only a small proportion of the genetic contribution to POAG risk, and although most of the reported associations have been validated in an independent study, only a few have been investigated in functional studies (e.g. CDKN2B-AS, SIX6, and CAV1/2).”

Further, we have added the references below to illustrate this point:

- Carnes MU et al. Discovery and functional annotation of SIX6 variants in primary open-angle glaucoma. PLoS Genet. 2014 (PMID: 24875647)
- Skowronska-Krawczyk D et al. P16INK4a Upregulation Mediated by SIX6 Defines Retinal Ganglion Cell Pathogenesis in Glaucoma. Mol Cell. 2015 (PMID: 26365380)
- Gao S, Jakobs TC. Mice Homozygous for a Deletion in the Glaucoma Susceptibility Locus INK4 Show Increased Vulnerability of Retinal Ganglion Cells to Elevated Intraocular Pressure. Am J Pathol. 2016 (PMID: 26883755)
- Elliott MH et al. Caveolin-1 modulates intraocular pressure: implications for caveolae mechanoprotection in glaucoma. Sci Rep. 2016 (PMID: 27841369)

2. The supplementary material should have a table of contents

We now provide a table of contents for the supplementary material.

3. **The rationale for the high genomic inflation factor is a little concerning. Upon review of the supplemental Figure 1, the high genomic inflation factor is driven by non-Hispanic whites. This raises concerns about cryptic relatedness in this subpopulation. The authors need to address how they minimized cryptic relatedness in GERA.**

Because of the large sample size and confirmed association signals, we obtained a genomic inflation factor lambda (λ) of 1.056 for the GERA multiethnic meta-analysis, which is reasonable for a dichotomous trait with polygenic inheritance with a sample size this large². Since λ scales with sample size, some have found it informative to report λ_{1000} ^{3,4}, the inflation factor for an equivalent study of 1000 cases and 1000 controls, which can be calculated by rescaling λ , as below:

$$\lambda_{1000} = 1 + (\lambda_{obs} - 1) * (1/n_{cases(1000)} + 1/n_{controls(1000)}) / (1/n_{cases(obs)} + 1/n_{controls(obs)})$$

, where $n_{cases(obs)}$ and $n_{controls(obs)}$ are the study sample size for cases and controls, respectively, and $n_{cases(1000)}$ and $n_{controls(1000)}$ are the target sample size (1000).

So, we have calculated the lambda 1000 for the non-Hispanic white sample, as follows:

$$\lambda_{1000} = 1 + (1.065 - 1) * (1/3,836 + 1/48,065) / (1/1000 + 1/1000)$$

$$\lambda_{1000} = 1.0091$$

We obtained the value of 1.0091 for λ_{1000} , which is reasonable for a genomic inflation factor under the assumption of polygenic inheritance.

Similarly, we have also calculated the lambda 1000 for the meta-analysis across the 4 race/ethnicity groups, and we have added this information in the Results section, and we have also added some references to justify the initial lambda value of 1.056, as below:

“Novel glaucoma loci in GERA

In our discovery GWAS analysis, we identified 12 independent genome-wide significant ($P < 5 \times 10^{-8}$) loci associated with POAG in the multiethnic meta-analysis ($\lambda = 1.056$, and $\lambda_{1000} = 1.006$, which is reasonable for a sample of this size under the assumption of polygenic inheritance²⁻⁴).”

4. **In the main results section, the authors should comment regarding the extent to which the new loci replicate in Hispanics, Asians and African people.**

This is an excellent suggestion. We have now described the results from Supplementary Table 2 showing the lead POAG SNPs ($P < 5 \times 10^{-8}$) by race/ethnicity group in the GERA discovery cohort, and added text in the Results section, as below:

“Novel glaucoma loci in GERA

In our discovery GWAS analysis, we identified 12 independent genome-wide significant ($P < 5 \times 10^{-8}$) loci associated with POAG in the multiethnic meta-analysis. Of the 12 loci, 5 were novel (41.7%), including rs56117902 in *FMNL2*, rs9494457 in *PDE7B*, rs149154973 near *ELN*, rs324794 near *TMTC2*, and rs2593221 in *TCF12*. We also examined the association of the lead SNPs at the 12 genome-wide significant loci with POAG in each individual race/ethnicity group (**Supplementary Table 2**). In African-Americans, SNPs rs56117902 at *FMNL2* and rs149154973 near *ELN* were both nominally associated with POAG ($P = 0.017$ and $P = 0.047$ for rs56117902 and rs149154973, respectively). In Hispanic/Latinos, we found a suggestive association between *PDE7B* rs9494457 and POAG risk ($P = 0.005$). In East Asians, we detected a nominal association of POAG with *TCF12* rs2593221 ($P = 0.019$). Except for rs149154973 near

ELN, all SNPs at novel loci showed consistent direction of effects across all race/ethnicity groups, and no significant heterogeneity was observed between race/ethnicity groups."

5. In line 159, the authors mention that they assess the lead 12 glaucoma SNPs in relation to IOP. They should clarify that this is IOP measured in GERA not UKB. In the methods section the authors should clarify what they mean by "valid IOP measurements". I assume this also means IOP as measured in cases and controls. How was IOP adjusted for cases that were on glaucoma treatment?

This is a good point, and we have now clarified in the Methods, as following:

In the Results:

"Secondary and sub-group analyses

To further investigate whether the POAG susceptibility loci identified in this study influence glaucoma susceptibility through their effect on IOP or independently of IOP, we conducted three additional analyses. First, we examined the association of lead SNPs at these loci with IOP, which was assessed in the GERA cohort."

Indeed, in our GERA study, IOP was measured in both POAG cases and controls.

In the Methods section, we have also clarified what we meant by "valid IOP measures", which refers to our recent paper (Choquet H, et al. Nat Commun. 2017 PMID: 29235454), as following:

"Case and control definition

All GERA subjects included in this study had valid IOP measures as previously described (Choquet H, et al. Nat Commun. 2017). Briefly, non-numeric entries for IOP, extreme values (≤ 5 and >60 mmHg), and measurements taken on a single eye were removed. Further, IOP measurements that were taken after initial prescription of IOP lowering medications were excluded to avoid values influenced by treatment".

6. On line 243, the authors state," IOP progressively crashes in some of the 244 B6.Lmx1bV265D/+ mice." This is vague jargon. Please explain exactly what this statement means.

We agree with the reviewer that the term "crashes" was inappropriate, and we now provide a more scientifically sounds description in the Results and Methods sections, as follows:

"The effect of Lmx1b mutations on glaucoma-related phenotypes depends on genetic background

Slit lamp-based clinical eye examination showed that the B6 mutant mice exhibit a severe developmental phenotype characterized by malformed eccentric pupils, irido-corneal strands, corneal haze and corneal scleralization (Figure 4.A.). The developmental phenotype on a 129 background was much milder, and mainly limited to mild pupillary abnormalities in about half of the mice. No major developmental abnormalities were detected in D2 mice with the *Lmx1b*^{Q82X} allele (focal corneal keratopathy is an unrelated strain characteristic of D2 mice that is frequently present in both wildtype and mutant mice). Further, with age, high IOP often results in a more open pupil configuration in D2.*Lmx1b*^{Q82X/+} mice. The B6.*Lmx1b*^{V265D/+} and D2.*Lmx1b*^{Q28X/+} mice were highly susceptible to developing glaucomatous nerve damage (Figure 4.B. and 4.C.), while 129.*Lmx1b*^{V265/+} mice rarely developed nerve damage (data not shown). Finally, *Lmx1b* mutations induced elevated IOP in all three strain backgrounds (Figure 4.D. and 4.E.). IOP is highly variable in the B6.*Lmx1b*^{V265D/+} which is possibly caused by a variety of reasons. One possible explanation for the spread of IOPs is due to *Lmx1b* mutants exhibiting abnormal corneas that are often stretched and sometimes perforate. This is a likely explanation, especially considering that B6 *Lmx1b* mutants have the most severely affected corneas as well as the lowest IOPs. Additionally, variability in IOP may result from stochastic variation in developmental consequences

resulting from the *Lmx1b* mutation. As precedents, various mutant genes that affect development of ocular drainage structures also variably result in maldevelopment of the ciliary body (Smith, 2000 & Chang, 2001)."

"IOP measurement

IOP was measured using the microneedle method as previously described in detail⁹⁰. Briefly, mice were acclimatized to the procedure room and anesthetized via an intraperitoneal injection of a mixture of ketamine (99 mg/kg; Ketlar, Parke-Davis, Paramus, NJ) and xylazine (9 mg/kg; Rompun, Phoenix Pharmaceutical, St. Joseph, MO) prior to IOP assessment - a procedure that does not alter IOP in the experimental window. All cohorts included male and female mice. The IOPs of B6 mice were assessed in parallel with experimental mice as a methodological control to ensure proper calibration and equipment function. The IOP values are highly variable in B6.*Lmx1b*^{V265D/+} eyes. *Lmx1b* mutants have abnormal corneas that are often stretched and sometimes perforate, and this perforation results in lower IOP values, which may explain the greater spread of IOP values in B6 *Lmx1b* mutants. To evaluate the change in the range of IOP values in *Lmx1b* mutant eyes across strains relative to controls, we used k-means clustering. We set k=3 for each individual group. The cluster with the highest IOP values was taken from each group and a one-way analysis of variance (ANOVA) was performed. Tukey's honest significance difference (HSD) was used to compare the means between *Lmx1b* mutant and WT mice within each strain."

7. Reference 22 is incomplete.

Reference 22 has now been completed, as below:

Choquet H, et al. A large multi-ethnic genome-wide association study identifies novel genetic loci for intraocular pressure. *Nat Commun.* 2017 Dec 13;8(1):2108. doi: 10.1038/s41467-017-01913-6. PMID: 29235454

8. The following statement is not up to date: "In fact, clinical trials to test the efficacy of ROCK inhibitors are underway to lower IOP in patients with glaucoma." The FDA in the US approved Netarsudil and Ripasudil is approved for clinical use in Japan.

We thank the reviewer for pointing out our oversight. We have revised this statement in the discussion, as below:

"Recently, an ophthalmic solution consisting of netarsudil 0.02% [Rhopressa[®]], a ROCK inhibitor has been approved in the U.S. for the reduction of elevated IOP in patients with POAG (Hoy SM. *Drugs* 2018 – PMID: 29453668). Another ophthalmic solution, ripasudil hydrochloride hydrate [Glanatec[®]], which is also a ROCK inhibitor, has been approved in Japan for the treatment of glaucoma and ocular hypertension (Garnock-Jones KP. *Drugs* 2014 – PMID: 25414122)."

Reviewer #4 (Remarks to the Author):

This manuscript describes a large body of work to identify genetic risk loci for primary open angle glaucoma (POAG). The authors begin with a GWAS meta-analysis of 4 ethnic groups from the GERA cohort, which is dominated by non-Hispanic whites, but also includes Hispanic/Latino, East Asian and African Americans. Replication of the findings are undertaken in the publicly available UK Biobank data. A full GWAS of UK biobank is also presented, with subsequent replication in the GERA cohort. Finally, a meta-analysis of GERA and UKB is presented, along with secondary analyses for IOP and conditional association. The authors have also explored the biology of the associated loci, evaluating actin stress fibre formation in a cell culture model, examining the glaucoma phenotype in a mouse mutant, and reporting the ocular tissue expression patterns in mouse and human of genes at the associated loci.

Overall, the authors report 9 novel loci for POAG with replication as well as 24 novel loci from the final meta-analysis, representing a substantial contribution to POAG genetics. Some additional information would clarify some sections.

Thank you to the reviewer for his/her positive comments and constructive review.

Comments for the authors

1. **The analysis of ancestry vs prevalence is really interesting, but I spent a long time trying to interpret Figure 1 due to scant methodological information on the generation of these plots. Please describe clearly how the prevalence is calculated for these plots. My assumption is that it was calculated per 'bin' of PC1 or PC2, but it is not at all clear**

We recognize that methodological information on how the plots of POAG prevalence vs. PCs across race/ethnicity groups have been generated was missing in the initial manuscript.

We have added text to the Methods section to provide information on the generation of these plots, as below:

"Plots of Primary Open-Angle Glaucoma Prevalence versus Genetic Ancestry

To visualize the glaucoma distribution by the ancestry PCs, we created a smoothed distribution of each individual i 's glaucoma phenotype using a radial kernel density estimate weighted on the distance to each other j^{th} individual, $\sum_j \varphi(\{d(i,j)/\max_{i',j'}[d(i',j')]*k\})$, where $\varphi(\cdot)$ is the standard normal density distribution, k is the smooth value (5 for non-Hispanic whites; and 15 for East Asians, Hispanic/Latinos and African-Americans), and $d(i',j')$ is the Euclidean distance based on the first two PCs. Race/ethnicity and/or nationality subgroup labels were derived from GERA or the Human Genome Diversity Project for visual representation of different groups (Banda Y et al. PMID: 26092716)."

We have also added some text in the Figure 1 Legend to be more informative, as follows:

"Figure 1: Plots of Primary Open-Angle Glaucoma Prevalence versus Genetic Ancestry in the Four GERA Race/Ethnicity Groups. POAG prevalence is indicated on a color scale, standardized across groups, with warmer colors indicating higher prevalence. Axes reflect the first two principal components of ancestry in each group. The phenotype distribution was smoothed over the PCs (within the individuals in each respective figure), which were divided by their standard deviation for interpretability (see Methods). Human Genome Diversity Panel populations are plotted at their relative positions in each figure. Human Genome Diversity Project populations are in a plain font, and GERA populations are in an italics font."

2. **Were outliers removed based on PCA in any of the cohorts? IF not, why not? The PCA plot for Hispanic/Latino appears to show individuals with strong African heritage. Were these individuals left in the Hispanic cohort or moved to the African American cohort based on the observed genetic ancestry?**

This is an important point to clarify. The classification of subjects in the PCA plots is based on self-reported race/ethnicity and not based on genetics, and no re-classification was done based on genetics, and no outliers were removed based on PCA. The rationale for doing this is that GERA participants who self-reported specific race/ethnicities have variable genetic contribution from different ancestral populations. For example, among Hispanic/Latinos, subjects who report Cuban or Puerto Rican race/ethnicity have a greater proportion of African ancestry than those that report Mexican race/ethnicity. The paper by Banda et al.⁶² provides a detailed description of the PCA and samples included. We have revised text in the Methods section to clarify this point.

“GWAS analysis and covariate adjustment

We first analyzed each of the four self-reported race/ethnicity groups (non-Hispanic whites, Hispanic/Latinos, East Asians, and African Americans) separately. We ran a logistic regression of POAG and each SNP using PLINK⁷¹ v1.9 (www.cog-genomics.org/plink/1.9/) with the following covariates: age, sex, and ancestry principal components (PCs). Data from each SNP were modeled using additive dosages to account for the uncertainty of imputation⁷².

Eigenstrat⁷³ v4.2 was used to calculate the PCs on each of the four race/ethnicity groups, and subjects were included for analyses in their self-reported group, as previously described⁶².

- 3. Table 1: shows NTG cases, which seems to relate to the analyses in sup Table 5, but what are the “OHTN controls” and where do they come into subsequent analyses? There is no dichotomised analysis of OHTN presented and the number seems very low if the definition is of IOP in the normal range.**

In our study, “OHTN (ocular hypertension) controls” are defined as participants who had a diagnosis of ocular hypertension (ICD-9 code, 365.04), but have no diagnosis of any type of glaucoma (any ICD-9 code, 365.xx other than 365.04). We have changed the row title in Table 1 to “Controls with OHTN diagnoses”, added a note following Table 1, and added a sentence in the Methods to reflect this point:

“Case and control definition

Subjects who had no diagnosis of any type of glaucoma (any ICD-9 code, 365.xx other than 365.04) but did have a diagnosis of ocular hypertension (OHTN) (ICD-9 code, 365.04), were included as controls.”

This sub-group of participants (N=881) is part of our control group for all analyses conducted in the GERA cohort and presented in our current paper: 1) GWAS of POAG (case/control analysis); 2) normal tension glaucoma (NTG) (743 NTG cases vs. 58,426 controls) and 3) high tension glaucoma (HTG) (4,243 HTG cases vs. 58,426 controls) sub-group analyses.

- 4. Supp Table 1: “GERA” covariate association with age, sex and the ancestry PCs”. Please confirm (in the table title) that this table is showing the association of each covariate (age, sex, ancestry PCs) with POAG in the GERA cohort. Also, please define how “Ashkenazi” is defined and where that data came from for the non-hispanic whites**

Yes, we confirmed that Supplementary Table 1 is showing the association of each covariate (age, sex, ancestry PCs) with POAG in the GERA cohort, and we have now modified the title of the table to reflect this point, as below:

“Supplementary Table 1. Association of each covariate (age, sex, ancestry PCs) with POAG in the GERA cohort.”

To clarify how Ashkenazi (ASHK) is defined, we have added the term “Ashkenazi ancestry proportion” to the footnote of the table and we now provide a description of the generation of this ancestry proportion in the Methods section, as follows:

“The ASHK proportion was extracted from the initial European principal components analysis, where individuals of European and Ashkenazi ancestries were run together to produce eigenvectors. The clusters resulting from this were re-classified as 0.0, 0.5, 0.75 and 1.0 ASHK (by drawing grids in the PC1-PC2 space). A full description of the ancestry analyses is provided in Banda et al. 2015⁶².”

5. The order of ethnicities changes between Figure 1 and Supp Figure 1. Please use the same order of sub-groups in all Tables and Figures to make it easier for the reader to follow

We have now reordered the race/ethnicity sub-groups (according to the sample size: highest to smallest) in all Tables and Figures for consistency (i.e. 1- Non-Hispanic whites, 2- Hispanic/Latinos, 3- East Asians, and 4- African-Americans).

6. Supp Figure 3: Please indicate the source data of the linkage disequilibrium and recombination rate on the LocusZoom plots.

To create those LocusZoom plots, we used “h19/1000 Genomes 2014 EUR” as the “Genome Build/LD Population”. We have added this information in the Supplementary Information file.

7. Table 2 (and other tables with Odds Ratios): Please provide the confidence intervals on the ORs

We now provide the 95% C.I. on the ORs in Table 2, as well as in Table 3, and in Supplementary Tables 2, 4-7 and 10.

8. Figure 2 and Supp Figure 4:

a. The differences in phalloidin staining between control and siRNA in the presence of FBS are not particularly convincing in Figure 2B. Supp Figure 4 more clearly shows reduced labelling in some cells. Note there is a typo in the figure legend (refers to Figure 1B instead of 2B). Please consider incorporating the Supp Figure images into the main figure to more clearly show the claimed result of altered actin stress filaments. The results for one of the two siRNAs could be moved to supplementary instead.

We thank the reviewer for suggesting structural changes to the figures. As recommended by the reviewer, in the revised manuscript we have incorporated the magnified images (previously shown in Supplementary Figure 4) into the main figure (originally Figure 2, now Figure 3) and moved the images of siRNA2 treated cells to Supplementary Figure 5. This way, the differences in phalloidin staining between control and siRNA in the presence of FBS shown in Figure 3 are more highlighted.

b. Also, which siRNA is used for the data presented in 2C? Finally, the conclusion of this paragraph (page 7, line 144) claims ‘reduced formation of actin stress fibres’. This may be the case, but the fibres have not been quantitated and the images don’t necessarily support this claim, depending which cell you look at. Please discuss this in more detail and alter the conclusion appropriately.

The data presented in the original Figure 2C (now Figure 3C) comes from experiment that used siRNA1. In the revised figure 3 (originally Figure 2), the data shown are generated using cells treated with siRNA1. The data generated using siRNA2 are shown in the supplemental figure 5 (originally Supplementary Figure 4).

As recommended by the reviewer, we have modified the text in the results by removing ‘reduced formation of actin stress fibers’. The revised text reads as:

“Overall, our data suggest that knockdown of *FMNL2* induces change in HTM cell morphology likely due to its effect on actin stress fiber assembly.”

c. The quantitation of the morphology seems more robust, although it is not clear how the statistics were handled from the methods. Is it a 2x3 chi-square test for the three morphology groups, or a 2x2, with a collapsing of groups?

This is a good point to clarify. For the experiment employing cells in absence of FBS (-FBS), we conducted a Chi-square test for a 2x2 table because we had two variables (round and intermediate cells). For the

10 minutes and 1 hour time points, we conducted a Chi-square test for a 2x3 table as there were three variables (round, intermediate and well-spread cells). In the Methods section, we have added statistical analysis information, as follows:

“We used 2x2 and 2x3 contingency tables for –FBS and +FBS conditions, respectively.”

d. Can “intermediate” and “well-spread” cells be indicated on the figure, to show the difference between these two morphologies?

In the revised figures, we have indicated the modestly spread cells (intermediate) with arrowheads and well-spread cells with asterisks.

e. Methods for this section, page 26 line 587: Were the cells serum starved immediately following transfection, or after 48 of growth? If the latter, was knockdown maintained through the 72 hours post-transfection?

The cells were first treated with siRNA for 24 hours. Followed by serum starvation (-FBS) for another 24 hours. We have now added this information in the Methods section, as below:

“Following *FMNL2* gene silencing using siRNAs for 24 hours, the HTM cells were serum-starved for another 24 hours (no FBS).”

9. Supp Table 5: How were the lead SNPs classified into high and normal IOP in this table and why? Is it based on association with IOP? Or NTG? Several loci in the ‘high IOP’ half of the table have negative associations with IOP, indicating the minor allele is associated with decreased IOP. While this means the other allele is associated with increased pressure, it is counter-intuitive with the labelling in this table and should be clarified. It would be easier to follow if the table columns matched the order of the description in the text on page 8 (i.e. IOP, NTG, HTG). The methods state that a GWAS was conducted for IOP, but only specific loci taken from the POAG GWAS are presented. Either present the full results, or modify the methods. Also, which statistic was used for the IOP analysis and were any covariates included?

We thank the reviewer for drawing our attention to the fact that in the initial manuscript we did not define what we meant by “High IOP loci” and “Normal IOP loci” regarding the header in Supplementary Table 5. We have now added a note, below this supplementary table, providing explanations, as below:

“Note: “High IOP loci” were defined as loci significantly associated with IOP (higher or lower) at a Bonferroni level of significance ($P < 0.0014$ for 12 SNPs in 3 analyses). In contrast, “Normal IOP loci” were defined as loci not associated with IOP (meaning with a $P \geq 0.0014$).”

As suggested by the Reviewer, we have also re-ordered the columns of this supplementary table, providing more consistency with the results text (i.e. IOP, NTG, and HTG).

Finally, we have now deleted the statement that a GWAS of IOP was conducted, and we modified and added text in the Methods, as follows:

“GWAS analysis and covariate adjustment

As a secondary analysis, we also assessed the associations between the 12 POAG-associated loci identified in GERA and IOP ... Individual’s mean IOP from both eyes for each visit was assessed, and the individual’s median of the mean across all the visits was used for analysis ... We ran a linear regression of IOP and each SNP using PLINK v1.9 with the following covariates: age, sex, and ancestry PCs.”

10. Supp Table 7: Why is the NCKAP5 locus not reported in the meta-analysis, even though the SNP is present in all 4 populations?

In our GERA sample, *NCKAP5* rs7588567 had an imputation quality score $r^2 < 0.8$ in non-Hispanic whites ($r^2 = 0.64$), in Hispanic/Latinos ($r^2=0.63$) and in African-Americans ($r^2=0.61$). For this reason, the result for the meta-analysis was not reported in Supplementary Table 7, as we consider it not fully reliable. We have added a note to reflect this point below Supplementary Table 7, as following:

“*NCKAP5* rs7588567 had a moderate imputation quality score r^2 in non-Hispanic whites ($r^2=0.64$), as well as in Hispanic/Latinos ($r^2=0.63$) and in African-Americans ($r^2=0.61$). For this reason, rs7588567 was excluded from the meta-analysis across the 4 race/ethnicity groups.”

11. Does the lead SNP at the *FMNL2*, or one in strong LD with it, influence the expression of the *FMNL2* gene, or any other nearby gene? These data can be accessed from the GTex project and could be informative for putative functional SNPs at all replicated loci, albeit, with the caveat of the specific tissues available in GTex.

This is an excellent suggestion. We have now tested whether the lead SNP rs56117902 at *FMNL2*, and an additional seven SNPs in strong LD with rs56117902 ($R^2>0.80$) had a significant GTex eQTL (<https://www.gtexportal.org/home/>).

No significant eQTLs were found for SNP rs56117902 in any tissue. However, 4 SNPs in strong LD with rs56117902 had a nominal significant GTex eQTL in thyroid tissue. Results are presented in the following table:

GTEx eQTLs for the *FMNL2* lead SNP and its proxy SNPs ($R^2>0.80$)

SNP	R^2	D'	GTEx - Single-Tissue eQTLs
rs56117902	-	-	No significant eQTLs
rs17399080	1.0	0.99	No significant eQTLs
rs11684450	1.0	0.94	No significant eQTLs
rs62180799	1.0	0.94	No significant eQTLs
rs4664586	0.95	0.85	2_153324071_C_T_b37 in ENSG00000213197.3 in thyroid ($P=0.000051$)
rs4664109	0.97	0.83	2_153294515_G_T_b37 in ENSG00000213197.3 in thyroid ($P=0.000025$)
rs6759772	0.94	0.81	2_153322724_C_T_b37 in ENSG00000213197.3 in thyroid ($P=0.000043$)
rs1878632	0.94	0.80	2_153323158_A_C_b37 in ENSG00000213197.3 in thyroid ($P=0.000043$)

Linkage disequilibrium (LD) metrics (R^2 and D') have all been calculated in European-ancestry populations using a web-based bioinformatic tool (<https://analysis-tools.nih.gov/LDlink/>).

12. Discussion page 15 line 349-352: please indicate here which genes specifically you are referring to, given that the data is in a supplementary figure. Are there any known pathway connections between these genes?

This is a good point. We now refer in the discussion to the genes that show consistent changes across stages in the RGC and optic nerve head, as follows:

“Here, we have identified a set of genes in POAG-associated loci whose expression is altered in the RGCs (e.g. *Ank*, *Cadm2*, *Six6*) and the optic nerve head (e.g. *Cadm2*, *Cdkn2b*) in two mouse models of glaucoma.”

To the best of our knowledge, there is no evidence for potential pathway connections.

13. The overall structure of the paper is quite confusing. The methods are results are ordered differently, making it very confusing to move between sections. The discussion is different

again. Please at least make the methods and results consistent. The results may be easier to follow if they present all the GWAS data, then the gene expression data for discovered loci, then the specific functional analyses of FMNL2 and Lmx1b.

This is a good suggestion. We have now restructured the manuscript according to the reviewers' suggestions to make the order consistent between the Results and Methods sections.

REFERENCES

- 1 Quigley, H. A. *et al.* The prevalence of glaucoma in a population-based study of Hispanic subjects: Proyecto VER. *Arch Ophthalmol* **119**, 1819-1826 (2001).
- 2 Yang, J. *et al.* Genomic inflation factors under polygenic inheritance. *Eur J Hum Genet* **19**, 807-812, doi:10.1038/ejhg.2011.39 (2011).
- 3 de Bakker, P. I. *et al.* Practical aspects of imputation-driven meta-analysis of genome-wide association studies. *Hum Mol Genet* **17**, R122-128, doi:10.1093/hmg/ddn288 (2008).
- 4 Freedman, M. L. *et al.* Assessing the impact of population stratification on genetic association studies. *Nat Genet* **36**, 388-393, doi:10.1038/ng1333 (2004).

Reviewer #1 (Remarks to the Author):

The authors have done an admirable job adequately addressing all of the reviewer comments and concerns. They have significantly improved their manuscript.

Reviewer #3 (Remarks to the Author):

The manuscript is much improved. There are a few minor issues that should be addressed:

1. The authors should indicate that the use of NTM5 cells as opposed to TM cells to study FMNL2 function represents another limitation of the study.
2. The authors do not adequately address the concerns of how IOP was handled in treated patients. The assertion that "IOP lowering medications are always prescribed before IOP lowering surgical interventions, the IOP measurements included in our analysis are values taken prior to any surgical interventions" is not entirely correct. Some patients do have laser trabeculoplasty or even surgery before medicines are used (especially now that minimally invasive incisional surgery is gaining popularity) and there are randomized clinical trials to support the use of such approaches. What approaches, if any were taken to exclude patients with initial trabeculoplasty or incisional surgery from analysis? I honestly don't think there are large numbers of such patients but this issue should be addressed.
3. In the Lmx1b mutant mice, please explain what is meant by "Further, with age, high IOP often results in a more open pupil configuration in D2.Lmx1bQ82X/+ mice."

Reviewer #4 (Remarks to the Author):

The authors have addressed all my previous questions.

There is a typo at line 287 "corneal" should be "cornea"

In the introduction, the authors highlight that there has been very little gene discovery undertaken in people of African heritage in particular and they claim to set out to fill that gap in knowledge. The African component of GERA is small compared to the other ethnic groups. Can the authors discuss the power of the African American cohort in this context and comment on the contribution to the overall findings from this population. It is notable that GWAS results are not presented for each ethnicity independently, presumably because the non-hispanic white cohort is powered appropriately. With the claims made in the introduction, it is important to return to this point in the discussion.

Dear Dr. Trenkmann,

We would like to thank the reviewers for their very enthusiastic feedback, including that “The authors have done an admirable job adequately addressing all of the reviewer comments and concerns”, and for their additional comments. We have followed these suggestions, and made changes to the manuscript to address these comments. Below, we provide detailed responses addressing the individual comments of the reviewers. We hope that the revised manuscript is now acceptable for publication.

Reviewers' comments:

Reviewer #1 (Remarks to the Author):

The authors have done an admirable job adequately addressing all of the reviewer comments and concerns. They have significantly improved their manuscript.

We thank the reviewer for the very positive feedback.

Reviewer #3 (Remarks to the Author):

The manuscript is much improved. There are a few minor issues that should be addressed:

1. The authors should indicate that the use of NTM5 cells as opposed to TM cells to study FMNL2 function represents another limitation of the study.

As requested by the reviewer, we have now included this limitation in the Discussion, as follows:

“We recognize several potential limitations of our study ... Finally, although investigating all implicated genes in the current study would be of great interest, we restricted our functional investigations to two genes (*FMNL2* and *LMX1B*) because they were implicated in the 95% credible set, were novel loci, replicated in an independent cohort, and resources were available for functional characterization. Further, the use of the transformed human NTM5 cells as opposed to primary HTM cells to study FMNL2 function represents another limitation of the study.”

2. The authors do not adequately address the concerns of how IOP was handled in treated patients. The assertion that “IOP lowering medications are always prescribed before IOP lowering surgical interventions, the IOP measurements included in our analysis are values taken prior to any surgical interventions” is not entirely correct. Some patients do have laser trabeculoplasty or even surgery before medicines are used (especially now that minimally invasive incisional surgery is gaining popularity) and there are randomized clinical trials to support the use of such approaches. What approaches, if any were taken to exclude patients with initial trabeculoplasty or incisional surgery from analysis? I honestly don't think there are large numbers of such patients but this issue should be addressed.

We acknowledge that the assertion that “IOP lowering medications are always prescribed before IOP lowering surgical interventions, the IOP measurements included in our analysis are values taken prior to any surgical interventions” was overstated.

We reviewed our data on these surgical interventions in subjects included in our published GWAS of IOP. We evaluated the number of patients who had laser trabeculectomy and other IOP lowering surgical interventions (i.e. trabeculectomy, tube shunt procedures, etc.) in GERA. These patients were identified from procedure codes captured in the KPNC electronic health records (EHR) system. In GERA, we identified 604 individuals who had laser trabeculectomy or other IOP lowering surgical interventions. Among these 604 individuals, 599 (99.2%) had an IOP measurement taken prior to surgical intervention. Given that the subjects who had IOP lowering surgical interventions represent less than one percent of our sample, and the proportion of subjects who did not have a pre-intervention IOP measurement represent less than one percent of that subset, the IOP values that we used in our analysis are unlikely to be influenced by IOP lowering surgical interventions.

So, to be completely accurate, we have revised our statement on these procedures to reflect this point in the Methods, as below:

“Further, IOP measurements that were taken after initial prescription of IOP lowering medications were excluded to avoid values influenced by treatment. Because IOP lowering medications are almost always prescribed before IOP lowering surgical interventions (i.e. laser trabeculectomy, trabeculectomy, tube shunt procedures, etc.), we did not remove subjects who had these surgical interventions.”

3. In the *Lmx1b* mutant mice, please explain what is meant by “Further, with age, high IOP often results in a more open pupil configuration in D2.*Lmx1b*^{Q82X/+} mice.”

What we mean by “open pupil configuration” in D2.*Lmx1b*^{Q82X/+} mice is that the pupil is more dilated in mutants as compared to the control mice. We have modified this sentence to clarify, as below:

“Further, with age, high IOP often results in a more dilated pupil in D2.*Lmx1b*^{Q82X/+} mice.”

Reviewer #4 (Remarks to the Author):

The authors have addressed all my previous questions.

There is a typo at line 287 "corneal" should be "cornea"

We thank the reviewer for drawing our attention to this typo. We have now corrected this term.

In the introduction, the authors highlight that there has been very little gene discovery undertaken in people of African heritage in particular and they claim to set out to fill that gap in knowledge. The African component of GERA is small compared to the other ethnic groups. Can the authors discuss the power of the African American cohort in this context and comment on the contribution to the overall findings from this population. It is notable that GWAS results are not presented for each ethnicity independently,

presumably because on the non-hispanic white cohort is powered appropriately. With the claims made in the introduction, it is important to return to this point in the discussion.

We recognize that our African American sample is the smallest group compared to the other race/ethnicity groups, and we may have been underpowered to detect individual SNP associations with statistical significance. We have added text to reflect this limitation in the Discussion, as below:

“We recognize several potential limitations of our study ... Third, in our study, we note that the African American subgroup has the smallest sample size compared to the other race/ethnicity groups, potentially limiting statistical power to detect some SNP associations. We note, however, that we did observe nominally significant associations with several previously reported and newly identified POAG risk loci in this group.”

Reviewer #3 (Remarks to the Author):

My comments have been addressed.

I thank the authors for addressing my concerns